# Round-trip Reinforcement Learning: Self-Consistent Training for Better Chemical LLMs

## Abstract

Large Language Models (LLMs) are emerging as versatile foundation models for computational chemistry, handling bidirectional tasks like reaction prediction and retrosynthesis. However, these models often lack round-trip consistency. For instance, a state-of-the-art chemical LLM may successfully caption a molecule, yet be unable to accurately reconstruct the original structure from its own generated text. This inconsistency suggests that models are learning unidirectional memorization rather than flexible mastery. Indeed, recent work has demonstrated a strong correlation between a model's round-trip consistency and its performance on the primary tasks. This strong correlation reframes consistency into a direct target for model improvement. We therefore introduce Round-Trip Reinforcement Learning (RTRL), a novel framework that trains a model to improve its consistency by using the success of a round-trip transformation as a reward signal. We further propose an iterative variant where forward and reverse mappings alternately train each other in a self-improvement loop, a process that is highly data-efficient and notably effective with the massive amount of unlabelled data common in chemistry. Experiments demonstrate that RTRL significantly **boosts performance and consistency** over strong baselines across supervised, self-supervised, and synthetic data regimes. This work shows that round-trip consistency is not just a desirable property but a trainable objective, offering a new path toward more robust and reliable foundation models.

## 1 Introduction

Large Language Models (LLMs) are emerging as a powerful class of versatile and generalizable foundation models for computational chemistry (Fang et al.; Cao et al., 2025; Wang et al., 2025). A key advantage is their ability to provide a more flexible and intuitive interface for scientists, enabling interaction with complex data through natural language. This is powered by their unique capability to process and generate information across diverse chemical modalities, from structured SMILES strings to unstructured experimental procedures (Zhao et al., 2025; Pei et al., 2024a). By unifying these disparate tasks and data types within a single framework, LLMs promise to significantly accelerate the pace of scientific discovery.

A prominent theme enabled by this new generation of models is the modeling of *bidirectional* chemical tasks, such as translating between textual descriptions and molecular structures or mapping between forward synthesis and retrosynthesis (Pei et al., 2024b; Edwards et al., 2024; Liu et al., 2024a). More than just a capability, however, this bidirectionality allows us to examine a model's depth of understanding through the crucial property of **round-trip consistency** (RTC). A model exhibits RTC if, for instance, after generating a textual caption for a given molecule, it can then take that caption as input and accurately reconstruct the original molecular structure. A failure to pass this test demonstrates a critical weakness: it relies on shallow, uni-directional memorization of statistical patterns rather than developing a flexible and abstract mastery of the underlying chemical principles. This distinction between memorization and true understanding provides critical insight into the strong positive correlation observed in recent work between a model's RTC and its overall task performance (Liu et al., 2024a; Allamanis et al., 2024). A consistent model, by necessity, must learn the fundamental rules and relationships governing the data. For example, a model that truly

understands an underlying reaction mechanism can reason bidirectionally, leading to both stronger round-trip consistency and superior generalization compared to a model that has only memorized common, unidirectional pairs. Ultimately, this deeper, more principled understanding is a prerequisite for scientific adoption, as high-stakes applications like therapeutic design demand models with exceptional overall knowledge utilization, reliability, and logical coherence.

While the importance of this property has been recognized in various domains (Rennie et al., 2020; Somers, 2005), existing work has primarily leveraged it in two ways, one as a post-hoc evaluation metric (Rennie et al., 2020) and the other as a data augmentation technique (Alberti et al., 2019). Consequently, a method that can **algorithmically** enforce and **iteratively** improve round-trip consistency directly within the model's training loop has remained a significant, unaddressed challenge.

To address this gap, we introduce **Round-Trip Reinforcement Learning** (RTRL), a novel framework designed to directly optimize for round-trip consistency. Our approach frames the problem as a self-supervised task where a forward model is trained to produce outputs that a backward model can successfully map back to the original input. **The success of this reverse mapping serves as a reward signal to improve the forward model.** The core insight is that this process compels the model to build a deeper and more coherent internal representation of the chemical world. Rather than learning a shallow, unidirectional statistical mapping, the model must learn the underlying, bidirectional relationship that connects entities. For example, consider a case where the forward model generates an ambiguous output $B$ from an input $A$. The reverse model, attempting to reconstruct the input, may then produce $A'$ where $A' \neq A$. RTRL uses the resulting low reward to penalize the initial $A \rightarrow B$ generation. To maximize its reward, the forward model must learn to discard ambiguous outputs and instead produce a clearer, more concrete representation $B^*$ from which the original input $A$ is recoverable. We further propose an iterative variant of RTRL where the forward and backward functions **swap roles** to continuously solidify the model's knowledge in a self-improving paradigm.

A significant advantage of the RTRL framework is its adaptability to different data availability. Primarily, the training process only requires access to data from a single domain (e.g., a list of known molecular products), without needing labelled pairs (e.g., their corresponding reactants), to improve the ability to generate correct responses (e.g., enhancing retrosynthesis ability by accessing only the products). Furthermore, this paradigm facilitates strong improvement over the base model in cases where data are either *supervised, self-supervised, or synthetic*. In our empirical evaluation, RTRL boosts self-consistency in terms of exactly by up to 52%, improves primary task performance by up to 55%. We also show the continuously improved efficacy of iterative LLM via self-play. These results validate our central thesis: that enforcing round-trip consistency is a powerful mechanism for unlocking its latent knowledge, leading to more robust and credible chemical foundation models.

## 2 PRELIMINARY

**Molecule data and tasks:** In this paper, we primarily work with data from the chemical domain, apart from regular chemical text, the molecules are represeted in Simplified Molecular Input Line Entry System (SMILES) sequence, it provides 1-D unambiguous representation of a molecule, which is a good fit for LLM processing. SMILES can also represents a Reaction in the format of "*Reactants>Reagents>Products*", where *Reactants*, *Reagents* and *Products* are chemicals in SMILES separated by dots. Concrete examples of SMILES and Reaction SMILES can be found in Appendix A. Note that unlike a traditional reaction equation, where the LHS and RHS of the equation must have the same elements, Reaction SMILES can omit secondary products and reactants to focus on the core transformation, so the elements in reactants and products might not match exactly.

This paper focuses on two pairs of bidirectional tasks, molecule captioning (molecule to description) verus text-based molecule generation (description to molecule) and reaction prediction (reactants to products) versus retrosynthesis (products to reactants). Modern chemical language models usually are equipped with all four functions (Fang et al.; Zhao et al., 2025; Pei et al., 2024a), as jointly training them potentially results in positive knowledge transfer.

**Group Relative Policy Optimization (GRPO):** GRPO (Shao et al., 2024) is a RL algorithm that finetunes a policy LLM without a explicit value function. For a prompt $t$, the policy LLM generates

a group of responses $G$, and the advantage for each generation in the group is computed as the normalized advantage over the average group performance:

$$\hat{A}_i = \frac{r_i - mean(r_1, ..., r_{|G|})}{std(r_1, ..., r_{|G|}) + \epsilon_{norm}} \quad \forall i \in G, \tag{1}$$

where $\epsilon_{norm}$ is a small value for numerical stability and $r_i$ is the scaler reward for each sample, and the advantages are used as signal to guide the policy update. GRPO uses a trust-region objective to optimize the model, the details can be found in Appendix E.

## 3 ROUND-TRIP CONSISTENCY IS CRITICAL IN BIDIRECTIONAL SYSTEMS

The core principle in this paper is round-trip consistency (RTC) (Somers, 2005; Yung et al., 2025). *This refers to a system's ability to take an input, map it to another domain, and then reliably reconstruct the original input from that output.* The ability to maintain this consistency is a crucial benchmark for a model's depth of understanding. Failures in this area often suggest that a model is

Table 1: Round-trip Consistency

| Task | Exact Match ↑ |
| --- | --- |
| Retro.→React. Pred. | 0.032 |
| React. Pred.→Retro. | 0.547 |
| Captioning→Generation | 0.170 |

relying on shallow memorization rather than developing a high-level understanding of the underlying chemical knowledge (Alberti et al., 2019; Hong et al., 2025).

Recent work has demonstrated that a model's overall performance is often **highly correlated with its consistency**; models demonstrating round-trip consistency tend to be more accurate and reliable (Liu et al., 2024a; Allamanis et al., 2024; Hong et al., 2025). This raises a critical question: *How consistent are current chemical LLMs?* We present a case study on an SOTA chemical LLM, ChemDFM Zhao et al. (2025). We conducted three round-trip experiments where we start with a molecule (molecules), perform forward and then backward transformation using the LLM, and eventually test if the output matches the original input exactly. In Table 1, we can see that even the most recent LLM with broad chemical knowledge struggles to output round-trip consistent results, revealing a significant consistency gap. This empirical evidence confirms that the consistency gap is not a minor issue but a systematic limitation in current models. Meanwhile, this also highlights a major opportunity: directly addressing this fundamental weakness could be a key to elevating the capabilities of chemical LLMs to the next level. And in this paper, we will introduce Round-trip Reinforcement Learning (RTRL) to do just that.

## 4 ROUND-TRIP REINFORCEMENT LEARNING

In a supervised learning setting, the objective is to learn a mapping from an input domain $\mathcal{X}$ to an output domain $\mathcal{Y}$. Given a dataset of paired examples $(x_i, y_i)$, the goal is to learn the parameters $\theta$ of a model $f : \mathcal{X} \to \mathcal{Y}$ that minimizes a predefined loss function, $\mathcal{L}$. The objective is typically expressed as:

$$\theta^* = arg \min_{\theta} \mathbb{E}_{(x,y) \sim \mathcal{D}}[\mathcal{L}(f_\theta(x), y)] \quad x_i \in \mathcal{X}, y_i \in \mathcal{Y} \tag{2}$$

In contrast to supervised learning, the principle of RTC provides a powerful **self-supervised objective**. This requires a system with bidirectional ability with a forward function $f$ and a corresponding backward function $g : \mathcal{Y} \to \mathcal{X}$. Then, a round-trip transformation can be defined as:

$$\text{Forward Pass:} y = f(x), \qquad \text{Backward Pass:} x' = g(y) = g(f(x)) \tag{3}$$

The system is considered consistent if the final output $x'$ is highly similar to the original input $x$. Therefore, the learning goal is to optimize the functions $f$ and $g$ to maximize their similarity. This can be formulated as maximizing the expectation of a similarity function $s(\cdot, \cdot)$:

$$\max_{f,g} \mathbb{E}_{x \sim \mathcal{X}}[s(x, g(f(x)))] \tag{4}$$

A key advantage of this formulation is that the objective can be optimized using **only inputs from** $\mathcal{X}$, without requiring corresponding ground-truth labels from $\mathcal{Y}$. This naturally facilitates self-supervised learning, and we will discuss this further in section 4.3.

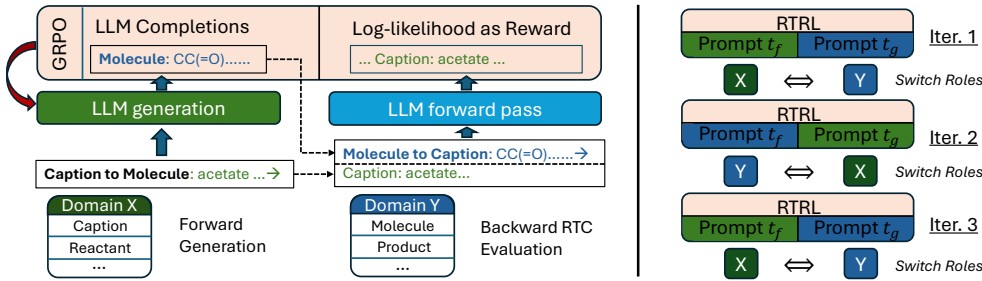

Figure 1: Left: Round-Trip Reinforcement Learning Pipeline. An input $x$ is mapped to an output $y$ by a forward prompt. We then compute the generation likelihood of $x$ given $y$ and a backward prompt. The likelihood is used as the reward for RL. Right: In iterative RTRL, we switch the forward and back prompts as well as the input and output domains to achieve mutual improvement.

This principle is particularly well-suited for Large Language Models (LLMs), where a single model can perform diverse functions conditioned on different prompts. The forward and backward functions, $f$ and $g$, are not separate models but are instantiations of the same LLM, parameterized by $\theta$, conditioned on a pair of inverse prompts, $t_f$ and $t_g$, and we can define

$$f(x) \equiv LLM_\theta(x, t_f), \quad g(y) \equiv LLM_\theta(y, t_f) \tag{5}$$

and we can rewrite the task to maximize RTC as,

$$\theta^* = \arg\max_\theta \mathbb{E}_{x \sim \mathcal{X}}[s(x, LLM_\theta(LLM_\theta(x, t_f), t_g))] \tag{6}$$

Optimizing this objective refines the model's internal parameters to ensure that the knowledge used for the forward and reverse tasks is coherent. Such a process encourages a shift from unidirectional memorization toward a more robust, bidirectional understanding.

## 4.1 IMPROVING ROUND-TRIP CONSISTENCY WITH REINFORCEMENT LEARNING

Optimizing Equation 6 is not straightforward, as the output of $LLM_\theta$ is a sampled text response, precluding the use of standard gradient-based methods. Hence, we propose to use Reinforcement Learning (RL) to train the system. While our framework is compatible with various policy gradient algorithms, we adopt Group-Relative Policy Gradient (GRPO) for its effectiveness in LLM training. The overall method pipeline is illustrated in Figure 1.

A well-defined reward function is critical for any RL system. An intuitive and natural choice is directly using the objective proposed in Equation 6. However, such a choice presents two significant challenges. First, it requires engineering a task-specific similarity function for the text sequence space, which may not capture true chemical similarity (e.g., the BLEU score is a poor metric to assess chemical similarity between SMILES). Second, it is computationally prohibitive, requiring two full autoregressive generation steps (forward and backward generation) for a single reward calculation. To handle the two problems, we propose a surrogate objective,

$$\theta^* = \arg\max_\theta p_\theta(x|LLM_\theta(x, t_f), t_g), \tag{7}$$

where $p_\theta(x|y)$ is the LLM's conditional probability of generating $x$, given prefix $y$. Instead of generating a full backward response, we use the conditional likelihood of the original input $x$ as the reward. This formulation handles the two problems mentioned above directly. Now, it uses the LLM itself as the judge to evaluate the similarity. With a higher likelihood to generate $x$, it increases the overall likelihood that the output is similar to the original input. Such an approach does not require manually setting up a similarity function, avoiding human effort and flawed similarity design. This is inspired by recent work in LLM-as-a-judge (Gu et al., 2025). Secondly, in this approach, the likelihood can be computed via a single forward pass, largely improving the overall efficiency.

Following this formulation, we use GRPO to optimize the objective. Specifically, we begin with two copies of the same LLM for the forward function $LLM_\theta$ and the backward function $LLM_\phi$, $\theta = \phi$ initially. The forward function is used as the policy model and will be updated by the GRPO

training process. The backward function computes the likelihood of the original output given the forward function output. In our preliminary study, we found that updating the parameter $\phi$ along with the policy parameter $\theta$ results in unstable reward and gradient during training, so we chose to only update the policy parameters $\theta$, while keeping $\phi$ fixed. So, the final objective for the RL training is,

$$\theta^* = \arg\max_\theta p_\phi(x|LLM_\theta(x, t_f), t_g). \tag{8}$$

Then, for a input $x$, we generate a group of output $Y$ using $LLM_\theta(x, t_f)$, and the reward for each generation is computed as,

$$r(y) = \log p_\phi(x|y, t_g) = \sum_{i=1}^{L} \log p_\phi(x_i|x_{<i}, y, t_g), \quad \forall y \in Y, \tag{9}$$

where $L$ is the length of the input, and we normalize the reward by $L$ in practice to balance the impact of different input lengths. Then, the group of rewards is passed to GRPO for model optimization.

For some special cases, the policy model can hack reward easily via replicating the input in the forward function, which results in exceptionally high reward, since LLM prunes to repeat itself. We can optionally add a format score to ensure the forward model generates expected output. For example, in a reaction prediction task, we use RDKit to compute the components in the output to ensure the output is a single chemical product. To summarize, the reward of an input $x$ and one of its generation $y$ is:

$$r(y) = \sum_{i=1}^{L} \log p_\phi(x_i|x_{<i}, y, t_g) + \alpha F(y), \tag{10}$$

where $F(\cdot)$ is an optional format reward function, and $\alpha$ is a hyperparameter to ensure the format score outweighs the likelihood reward, so any reward hacking behavior is harshly penalized. We provide a more detailed discussion on the format reward in Appendix F.

### 4.2 ITERATIVE RTRL FOR MUTUAL IMPROVEMENT

The framework described thus far details how Round-Trip Reinforcement Learning (RTRL) can enhance the consistency of a forward mapping, leveraging the dataset only from the input domain $\mathcal{X}$. This is a critical application in computational chemistry, where large databases of molecules, $X$, often exist without corresponding ground-truth labels, $Y$ (descriptions, synthetic pathways, etc.).

A natural extension of this framework is to consider a symbiotic relationship: *can the now-improved forward model be used to enhance its inverse counterpart?* Once the forward generator $LLM(\cdot, t_f)$ has been refined, its ability to produce high-quality outputs makes it a **more reliable judge** for the reverse task. A more accurate judge provides a more meaningful reward signal, which in turn facilitates a more effective training process for the reverse mapping, $g : \mathcal{Y} \to \mathcal{X}$. This creates a virtuous cycle where each model bootstraps the other.

To implement this reverse training, a dataset from the target domain, $Y \subset \mathcal{Y}$, is required. Critically, this dataset Y does not need to be paired with the original dataset $X$. The two can be independent collections of data, potentially covering different subdomains of the problem space. This flexibility allows our framework to operate in various data regimes, see section 4.3 for a detailed discussion.

The iterative training procedure alternates between forward and reverse optimization phases. Starting with a model with parameters $\theta_k$ at iteration $k$, we first perform the standard RTRL, optimizing the objective in Equation 8. In the next phase, the roles are reversed, and we optimize:

$$\theta_{k+1} = \arg\max_\theta p_{\phi_k}(y|LLM_\theta(y, t_g), t_f). \tag{11}$$

The prompts $t_f$ and $t_g$ are **switched**, and $\phi_k$ and $\theta$ are initialized by $\theta_k$. We can then repeat this process for a fixed number of iterations or until the model stops improving, progressively refining both mappings to be more accurate and mutually consistent.

### 4.3 DISCUSSION ON THE USE AND ADVANTAGE OF RTRL

The RTRL framework is highly adaptable, offering distinct advantages across various scenarios of data availability. This section outlines four key use cases, with empirical evaluations for each presented in Section 6.

The most direct application of RTRL is **enhancing a single mapping**, $f : \mathcal{X} \to \mathcal{Y}$, via enforcing RTC only on data from the source domain, $\mathcal{X}$. This is critical in fields like chemistry, where large databases of molecules ($\mathcal{X}$) exist without paired synthesis pathways ($\mathcal{Y}$), or in software engineering for migrating legacy codebases with limited examples of migrated code. By forcing the generator to produce outputs that its own inverse function can recognize, RTRL compels a deeper, more robust understanding of the forward task, even without ground-truth labels.

In our **iterative training scheme**, the forward and reverse models can co-evolve in an iterative manner. A key strength of this process is that the datasets for the forward and reverse tasks do not need to be paired. This allows the framework to leverage two independent, potentially disjoint collections of data to facilitate mutual improvement. This dramatically increases the flexibility of data collection. Section 6 provides a concrete example where RTRL can use these unpaired USPTO-50K and USPTO-Mixed datasets to synergistically improve performance on both tasks.

In a more traditional **supervised** setting where paired data $(X, Y)$ is available, RTRL serves as a strong regularizer. While standard supervised fine-tuning optimizes for accuracy on the primary task, RTRL introduces an additional objective that enforces logical consistency and reliability, which reduces overfitting and encourages the model to learn more robust and generalizable representations.

Finally, RTRL facilitates a **self-play** paradigm where the target dataset, $Y$, is generated **synthetically** by the model's own forward function from a seed dataset $X$. This removes the dependency on label collection. The RL process, with its inherent exploration, enables the model to probe its own understanding, search the problem domain with synthetic data, and reinforce consistent reasoning paths. The model can refine and uncover latent knowledge entirely without external supervision.

## 5 RELATED WORK

**Chemical Language Model:** Earlier efforts such as MolT5 (Cao et al., 2025) and BioT5 (Pei et al., 2024a) established language-to-molecule translation task. The surge of LLM leads to extensive efforts to adopt LLM to molecular learning and science discovery for its exceptional sequential modeling ability and generalizability (Wang et al., 2025; Yang et al., 2024; Cheng et al., 2021). Most works focus on collecting high-quality pre-training (Zhao et al., 2025; Dey et al., 2025; Ye et al., 2025; Xia et al., 2025; Liu et al., 2024c;b) and instruction-tuning (Cao et al., 2025; Fang et al.; Yu et al.; Zhang et al., 2024) datasets to enhance models' ability to comprehend both chemical entities (molecule, SMILES, IUPAC names) and content (chemical bio-medical texts). Apart from the data perspective, recent works align LLM with structural or chemical knowledge to enhance native LLM's performance on chemical tasks (Li et al., 2024; Lin et al., 2024; Guo et al., 2025). Concurrently, several works propose to use reinforcement learning or preference optimization to chemically align properties better (Jang et al., 2024; Calanzone et al., 2025; Gkoumas, 2024; Lee et al., 2025). Compared to existing works, we propose RTRL that only needs unlabelled data to significantly improve the performance, largely reducing the difficulty of training an effective chemical LLM.

**Self-improving and Reinforcement Learning:** LLM self-improving has been a popular topic in various domains. Mostly in math and coding generation, self-improving techniques have achieved considerable success due to tools available for verifiable reward generation in these domains (Gao et al., 2025; Ma et al., 2025; Huang et al., 2025; Guan et al.). Another line of work also facilitates unsupervised/self-supervised self-improving without ground-truth label (Huang et al., 2023; Agarwal et al., 2025; Shao et al., 2025). These works strive to increase a model's confidence during output, while RTRL intends to check whether the output is consistent with the model's knowledge, providing a new lens into self-improving, particularly in the chemistry domain.

## 6 EXPERIMENT

We strive to answer the following research questions: **Q1:** Does RTRL actually improve round-trip consistency? **Q2:** Does enforcing round-trip consistency improve the model performance in both self-supervised or supervised cases? **Q3:** Does iterative RTRL bring an extra performance boost, and when does the model stop improving? **Q4:** Can iteratively improve the model from a seed dataset? **Q5:** Does RTRL work for different base models? **Q6:** How does RTRL compare

Table 2: Round-trip consistency improvement by RTRL.

| Task / Model | BLEU ↑ | Lev. ↓ | Exact Match ↑ | MACCS SIM. ↑ | RDKit SIM. ↑ | Morgan SIM. ↑ | FCD ↓ | Validity ↑ |
|---|---|---|---|---|---|---|---|---|
| *USPTO-Mixed Retrosynthesis→Reaction Prediction* | | | | | | | | |
| ChemDFM | 0.535 | 59.355 | 0.032 | 0.692 | 0.662 | 0.499 | 30.318 | 0.975 |
| **RTRL+ChemDFM** | **0.535** | **45.550** | **0.033** | **0.718** | **0.705** | **0.531** | **2.615** | **0.986** |
| *USPTO-50 Reaction Prediction→Retrosynthesis* | | | | | | | | |
| ChemDFM | 0.791 | 12.752 | 0.547 | 0.885 | 0.836 | 0.809 | **19.987** | 0.980 |
| **RTRL+ChemDFM** | **0.924** | **4.186** | **0.821** | **0.958** | **0.945** | **0.932** | 20.723 | **0.990** |
| *CHEBI-20 Molecule Captioning→Text-based Molecule Generation* | | | | | | | | |
| ChemDFM | 0.598 | 43.747 | 0.170 | 0.723 | 0.591 | 0.504 | 28.474 | **0.983** |
| **RTRL+ChemDFM** | **0.693** | **31.831** | **0.239** | **0.795** | **0.668** | **0.598** | **1.724** | 0.962 |

Table 3: Self-supervised performance on molecule-based tasks compared to pre-trained LLMs.

| Task / Model | BLEU ↑ | Lev. ↓ | Exact Match ↑ | MACCS SIM. ↑ | RDKit SIM. ↑ | Morgan SIM. ↑ | FCD ↓ | Validity ↑ |
|---|---|---|---|---|---|---|---|---|
| *CHEBI-20 Text-based Molecule Generation* | | | | | | | | |
| GPT (0-shot) | 0.475 | 48.985 | 0.164 | 0.605 | 0.475 | 0.441 | 20.837 | 0.687 |
| Qwen-8B | 0.030 | 665.894 | 0.005 | 0.240 | 0.130 | 0.101 | 29.719 | 0.468 |
| Mol-Instruction | 0.601 | 42.204 | 0.144 | 0.764 | 0.573 | 0.477 | 6.700 | **0.999** |
| ChemDFM | 0.846 | 15.652 | 0.546 | 0.898 | 0.779 | 0.733 | **2.088** | 0.978 |
| **RTRL+ChemDFM** | **0.852** | **15.022** | **0.553** | **0.901** | **0.781** | **0.735** | 2.099 | 0.982 |
| *USPTO-Mixed Reaction Prediction* | | | | | | | | |
| GPT (0-shot) | 0.691 | 16.194 | 0.433 | 0.819 | 0.768 | 0.727 | 5.968 | 0.910 |
| Qwen-8B | 0.231 | 62.644 | 0.005 | 0.368 | 0.271 | 0.251 | 21.436 | 0.942 |
| Mol-Instruction | 0.307 | 28.725 | 0.096 | 0.578 | 0.436 | 0.385 | 4.193 | **0.999** |
| ChemDFM | 0.845 | 8.685 | 0.559 | 0.880 | 0.831 | 0.803 | 18.818 | 0.987 |
| **RTRL+ChemDFM** | **0.857** | **7.934** | **0.601** | **0.895** | **0.851** | **0.823** | **0.171** | 0.985 |
| *USPTO-50K Retrosynthesis* | | | | | | | | |
| GPT (0-shot) | 0.601 | **27.388** | 0.104 | 0.609 | 0.464 | 0.441 | 14.624 | 0.821 |
| Qwen-8B | 0.571 | 35.589 | 0.000 | 0.638 | 0.546 | 0.519 | 22.522 | 0.962 |
| Mol-Instruction | 0.370 | 33.138 | **0.202** | 0.779 | 0.641 | 0.601 | **8.264** | **1.000** |
| ChemDFM | 0.583 | 36.762 | 0.171 | 0.796 | 0.738 | 0.650 | 22.347 | 0.988 |
| **RTRL+ChemDFM** | **0.625** | 31.996 | 0.151 | **0.805** | **0.759** | **0.671** | 21.449 | 0.992 |

to methods that can also do self-supervised learning, such as entropy minimization (Agarwal et al., 2025) and round-trip sample augmentation (Tetko et al., 2020)? We answer **Q1-4** in this section, and answer **Q5-6** in Appendix C. We also include qualitative examples in Appendix D. Details for experiments, codes, and reproduction can be found in Appendix B.

**Datasets and evaluations:** For molecule captioning and text-based molecule generation, we use CHEBI-20 (Edwards et al., 2021) and Language-Plus-Molecule-24 (LM-24) (Edwards et al., 2024). For reaction prediction and retrosynthesis, we use USPTO-Mixed (Jin et al., 2017) and USPTO-50K datasets (Schneider et al., 2016), respectively. Detailed description of each dataset can be found in Appendix A. To evaluate text, we use conventional NLP metrics including BLEU score, ROUGE scores, and METEOR scores. To evaluate molecules, we follow the convention in existing chemical LLM work (Edwards et al., 2022) to compare the exact match, validity, fingerprint similarity (MACCS, RDKit, Morgan), translation similarity (BLEU, Levenshtein distance), and Fréchet ChemNet Distance (FCD) (Preuer et al., 2018). In our result table, we use upward (downward) arrows to indicate that the metric is higher (lower) the better.

**Baselines:** We use Close-sourced LLM (GPT (OpenAI et al., 2024)), open-weight LLM (Qwen3-8B (Yang et al., 2025)), strong chemical LLMs (Mol-Instruction (Fang et al.) and ChemDFM (Zhao et al., 2025)) as our baselines to cover a wide spectrum of chemically-capable LLMs. Both chemical LLMs use Llama-3-8B series as the base model, and use a large chemical corpus to enhance the model's chemical ability. In this section, we use ChemDFM as the base model for RTRL.

**Round-trip consistency:** To answer **Q1**, we train ChemDFM with RTRL using only one domain of data (the data are unlabelled), and we compare the model after RTRL with ChemDFM. After training, the consistency evaluation is performed on the test dataset. The results are in Table 2. We see improved round-trip consistency on all round-trip generation tasks. In particular, we see 43% and 52% relative exact match improvement in molecule captioning to generation and reaction prediction to retrosynthesis tasks. This validates our design to use an RL to improve the base model's RTC in bidirectional tasks **with only data from one domain** and **using a base model's own knowledge**. RTRL allows one to use an unlabelled molecule dataset to improve the model, which is a prevalent scenario in the chemical domain.

Table 4: RTRL performance on text-based tasks.

| Task / Model | BLEU-2 ↑ | BLEU-4 ↑ | ROUGE-1 ↑ | ROUGE-2 ↑ | ROUGE-L ↑ | METEOR ↑ |
|---|---|---|---|---|---|---|
| *CHEBI-20* | | | | | | |
| GPT (0-shot) | 0.026 | 0.007 | 0.086 | 0.023 | 0.055 | 0.173 |
| Qwen-8B | 0.019 | 0.004 | 0.099 | 0.025 | 0.070 | 0.138 |
| Mol-Instruction | 0.076 | 0.056 | 0.249 | 0.175 | 0.236 | 0.164 |
| ChemDFM | 0.286 | 0.244 | 0.406 | 0.312 | 0.378 | 0.345 |
| **RTRL+ChemDFM** | **0.447** | **0.380** | **0.529** | **0.406** | **0.483** | **0.481** |
| *CHEBI-20 Supervised* | | | | | | |
| GPT (10-shot) | 0.042 | 0.015 | 0.131 | 0.040 | 0.089 | 0.231 |
| Qwen-8B | 0.163 | 0.098 | 0.398 | 0.219 | 0.341 | 0.325 |
| Mol-Instruction | 0.434 | 0.329 | 0.529 | 0.358 | 0.467 | 0.466 |
| ChemDFM | 0.385 | 0.333 | 0.493 | 0.388 | 0.457 | 0.437 |
| GRPO+ChemDFM | 0.418 | 0.358 | 0.523 | 0.397 | 0.475 | 0.457 |
| **RTRL+ChemDFM** | **0.492** | **0.421** | **0.563** | **0.437** | **0.515** | **0.521** |
| *LM-24 Supervised* | | | | | | |
| GPT (10-shot) | 0.013 | 0.004 | 0.056 | 0.018 | 0.040 | 0.114 |
| Qwen-8B | 0.745 | 0.537 | 0.763 | 0.572 | 0.550 | 0.706 |
| Mol-Instruction | 0.764 | **0.554** | 0.782 | 0.589 | 0.562 | 0.727 |
| ChemDFM | 0.752 | 0.545 | 0.779 | 0.586 | 0.562 | 0.720 |
| GRPO+ChemDFM | 0.734 | 0.534 | 0.748 | 0.555 | 0.548 | 0.708 |
| **RTRL+ChemDFM** | **0.766** | 0.551 | **0.790** | **0.599** | **0.567** | **0.735** |

Table 5: Supervised performance on molecule-based tasks compared to fine-tuned LLMs.

| Task / Model | BLEU ↑ | Lev. ↓ | Exact Match ↑ | MACCS SIM. ↑ | RDKit SIM. ↑ | Morgan SIM. ↑ | FCD ↓ | Validity ↑ |
|---|---|---|---|---|---|---|---|---|
| *LM-24 Text-based Molecule Generation* | | | | | | | | |
| GPT (10-shot) | 0.595 | 44.179 | 0.000 | 0.608 | 0.493 | 0.404 | 17.065 | 0.925 |
| Qwen-8B | 0.277 | 207.726 | 0.000 | 0.681 | 0.623 | 0.466 | 73.977 | 0.857 |
| Mol-Instruction | 0.411 | 120.475 | 0.000 | 0.696 | 0.637 | 0.476 | 26.684 | **1.000** |
| ChemDFM | 0.688 | 49.169 | **0.001** | 0.740 | 0.669 | 0.493 | 63.314 | 0.969 |
| GRPO+ChemDFM | 0.689 | 49.118 | 0.000 | 0.739 | 0.673 | 0.492 | 62.019 | 0.976 |
| **RTRL+ChemDFM** | **0.737** | **39.948** | **0.001** | **0.762** | **0.683** | **0.524** | **3.307** | 0.996 |
| *CHEBI-20 Text-based Molecule Generation* | | | | | | | | |
| GPT (10-shot) | 0.529 | 42.030 | 0.119 | 0.632 | 0.517 | 0.454 | 21.707 | 0.731 |
| Qwen-8B | 0.162 | 221.933 | 0.009 | 0.524 | 0.348 | 0.277 | 24.704 | 0.830 |
| Mol-Instruction | 0.479 | 68.238 | 0.083 | 0.714 | 0.499 | 0.406 | 2.647 | **0.993** |
| ChemDFM | 0.821 | 18.712 | 0.494 | 0.887 | 0.804 | 0.757 | 2.087 | 0.958 |
| GRPO+ChemDFM | 0.811 | 19.690 | 0.504 | 0.872 | 0.793 | 0.758 | 2.034 | 0.968 |
| **RTRL+ChemDFM** | **0.868** | **14.311** | **0.547** | **0.904** | **0.815** | **0.772** | 1.372 | 0.984 |
| *USPTO-Mixed Reaction Prediction* | | | | | | | | |
| GPT (10-shot) | 0.685 | 16.269 | 0.418 | 0.790 | 0.746 | 0.702 | 6.785 | 0.881 |
| Qwen-8B | 0.662 | 20.260 | 0.031 | 0.653 | 0.583 | 0.507 | 18.346 | 0.931 |
| Mol-Instruction | 0.660 | 19.135 | 0.011 | 0.653 | 0.563 | 0.490 | 8.470 | **0.999** |
| ChemDFM | 0.891 | 6.433 | 0.660 | 0.908 | 0.865 | 0.846 | 0.115 | 0.989 |
| GRPO+ChemDFM | 0.892 | 6.451 | 0.650 | 0.909 | 0.866 | 0.845 | 0.125 | 0.994 |
| **RTRL+ChemDFM** | **0.896** | **6.358** | **0.665** | **0.909** | **0.868** | **0.848** | **0.115** | 0.985 |
| *USPTO-50K Retrosynthesis* | | | | | | | | |
| GPT (10-shot) | 0.674 | 24.179 | 0.104 | 0.687 | 0.524 | 0.513 | 13.533 | 0.896 |
| Qwen-8B | 0.486 | 46.429 | 0.015 | 0.739 | 0.663 | 0.546 | 23.224 | 0.944 |
| Mol-Instruction | 0.588 | 36.660 | 0.004 | 0.697 | 0.540 | 0.474 | 12.352 | **0.999** |
| ChemDFM | 0.802 | 17.249 | 0.311 | 0.840 | 0.766 | 0.728 | 0.293 | 0.990 |
| GRPO+ChemDFM | 0.803 | 17.242 | 0.311 | 0.843 | 0.769 | 0.730 | **0.288** | 0.992 |
| **RTRL+ChemDFM** | **0.810** | **17.158** | **0.323** | **0.848** | **0.778** | **0.738** | 0.301 | 0.990 |

**Round-trip consistency and model performance:** Simply maximizing consistency is not suffi-
cient. Hence, we need to answer **Q2** by studying the performance of the RTRL-trained models.
In the self-supervised scenario, we only have access to one domain of data. The molecule-based
results are in Table 3, and the text-based results are in Table 4. We can see the RTRL-trained model
outperform the base ChemDFM on almost every task. RTRL achieved 7.5% relative improvement
in exact match on the reaction prediction task (0.601 vs 0.559). It also achieves at least 27% relative
improvement in all metrics on the CHEBI-20 molecule captioning task. These findings confirm our
central goal to use RTC to improve the model's coherence and performance. Meanwhile, we also
observe that the model has a slightly worse exact match for the retrosynthesis task on the USPTO-50
dataset. We found that after RTRL, our model tends to generate the full reaction SMILES, whereas
the USPTO-50 test set preferred core-reaction-focused equation as discussed in Section 2. How-
ever, from other metrics like fingerprint similarity, we can see that our model can generate more
chemically-correct outputs (See Appendix D for examples).

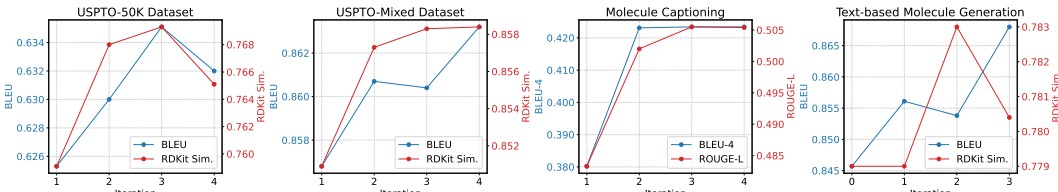

Figure 2: Model performance progression in iterative RTRL. From left to right: (1) Retrosynthesis performance. Supervised. (2) Reaction prediction. Supervised. (3) CHEBI-20. Synthetic. (4) CHEBI-20. Synthetic. Iteration 0 means the base model.

In the supervised setting, we perform an SFT on the bidirectional tasks before RL. During RL training, we add the evaluation metric score as an auxiliary score to fully utilize the label. We use (a subset of) the finetuning data in RL to ensure that the benefit does come from extra data. For the baselines, we compared the same set of LLMs as in the self-supervised scenario, but they are also finetuned with the same data. We additionally compared to GRPO+ChemDFM, where only the metric scores are used as a reward in the GRPO. The molecule-based results are in Table 5, and the text-based results are in Table 3. In this scenario, we still see improvement on most tasks. We also include model performance when RTRL is applied to another strong baseline Llasmol (Yu et al.) in Appendix C. RTRL brings performance improvement to a different base model, showing that RTRL is a general enhancer.

While RTRL is primarily designed to handle generative tasks, we show that RTRL can improve a model's inherent chemical knowledge, and is useful for classification and untrained chemical tasks. See Appendix C for details.

**Iterative RTRL:** To answer **Q3**, we apply RTRL to USPTO-50 and USPTO-Mixed dataset. They are a pair of bidirectional tasks, but their data entries do not match. We apply RTRL iteratively by starting with the reaction prediction task and alternating with retrosynthesis. We repeat the process 4 times and record the model performance on both tasks at different timestamps. We draw a line plot of BLEU and RDKit Similarity in Figure 2. We see that iterative RL can continuously improve the model's ability in both directions. This greatly extends the applicable scenario and effectiveness of RTRL to accommodate different data availability. We also note that some metrics show degradation after a certain number of iterations. We suspect that this is primarily because RTRL has uncovered most of the hidden knowledge in the pretrained chemical LLM and has exhausted the benefit of the two datasets, and the model begins to overfit to the training data.

**Synthetic RTRL:** To answer **Q4**, we apply RTRL to CHEBI-20 with caption removed as the seed dataset. Then we swap the roles of forward and backward functions and use the improved forward function to generate synthetic data from the seed dataset. Then, we use RTRL on this synthetic data to train the forward (previously backward) function. We repeat this process 3 times and plot performance on both directions in Figure 2. We can see clear improvement for both directions even without a ground-truth label from one domain. This indicates both the high quality of synthesized samples and the exceptional ability of LLM that can be uncovered by RTRL. Meanwhile, the improvement is less consistent compared to the non-synthetic case, potentially because of the highly variable process, as the performance not only depends on training, but also on data sampling. We leave designing a more controlled synthesis process to future work.

# 7 CONCLUSION, LIMITATION, AND FUTURE WORK

In this paper, we raised awareness of algorithmically enforcing RTC in LLM, which greatly contributes to an LLM's coherence and performance. We then propose a novel algorithm, RTRL, that can effectively inject such consistency into LLM using the GRPO algorithm. Empirical results show significant and consistent improvement over the base model, validating the algorithm's high efficacy. While RTRL is highly generic, it falls short on completely synthetic scenarios, where the seed dataset also needs to be synthesized. In the future, we plan to resolve this obstacle. We can then view any problem as a bidirectional mapping between question and answer and take RTRL as a principle to improve the model, and extend RTRL to broader domains such as code generation and math problem solving.

## REPRODUCIBILITY STATEMENT

To ensure the reproducibility of the paper, we included source code along with detailed instruction to reproduce the experimental results as supplementary material as well anonymous link attached in Appendix B. We also included experiments setup, hyperparameters, and dataset details in Appendix B. We disclosed the hardware used to train the model and to do inference. All datasets are accesible through online resources, and the access instruction are provided in the attached source code.

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

APPENDIX

# A  DATASET DETAILS

This section provides a detailed description about each dataset used in the experiments. Examples of each dataset and task can be found in Table 6.

Table 6: Examples of data and tasks.

| Dataset | Task | Input | Output |
|---------|------|-------|--------|
| CHEBI-20 | Molecule Captioning | COc1cccc2[nH]cc(C/C(=N/OS(=O)(=O)[O-])S[C@@H]3O[C@H](CO)[C@@H](O)[C@H](O)[C@H]3O)c12 | The molecule is an indolylmethylglucosinolate that is the conjugate base of 4-methoxyglucobrassicin, obtained by deprotonation of the sulfo group. It is a conjugate base of a 4-methoxyglucobrassicin. |
| CHEBI-20 | Text-based Molecule Generation | The molecule is a 1,3-thiazolium cation that is 1,3-thiazol-3-ium substituted by a methyl group at position 4, a (4-amino-2-methylpyrimidin-5-yl)methyl group at position 3, a 3-carboxy-1-hydroxypropyl at position 2 and a 2-[hydroxy(phosphonooxy)phosphoryl]oxyethyl group at position 5. It has a role as a human metabolite and a mouse metabolite. | Cc1ncc(C[n+]2c(C(O)CCC(=O)O)sc(CCOP(=O)(O)OP(=O)(O)O)c2C)c(=N)[nH]1 |
| LM-24 | Molecule Captioning | CCCCCCCCCCCCCCCCCCCCCC(=O)OC[C@H](COC(=O)CCCCCCCCCCCCCCCCCC)OC(=O)CCCCCCCCC(C)C | The molecule is a energy storage, nutrient, membrane stabilizer that impacts cardiovascular disease, metabolic syndrome, and pancreatitis. The molecule is a member of the thyroxine treatment class and affects both atherosclerosis and cancer. The molecule is a fat storage, a energy source, and a inflammatory, and it impacts obesity. |
| LM-24 | Text-based Molecule Generation | When heated to decomposition it emits acrid smoke and irritating fumes. The molecule has both a Bitter and unpleasant taste and a Pleasant odor. | COc1ccc(C(C)=O)cc1 |
| USPTO-Mixed | Reaction Prediction | CC#N.CCN(CC)CC.COC(=O)C1(OC)CC(C)C(=O)C=C1O.Cc1nc(C(F)(F)F)ccc1C(=O)Cl.N#[C][K] | COC(=O)C1(OC)CC(C)C(=O)C(C(=O)c2ccc(C(F)(F)F)nc2C)=C1O |
| USPTO-50K | Retrosynthesis | CNC(=O)c1ccc2c(c1)CC(=O)N2C1CCN(CC(=O)N2C[C@H]3CCC[C@H]3C2)CC1 | CNC(=O)c1ccc2c(c1)CC(=O)N2C1CCNCC1.O=C(CCl)N1C[C@H]2CCC[C@H]2C1 |

**CHEBI-20 Dataset:** The ChEBI-20 dataset, as prepared for the paper Edwards et al. (2021), is a cross-modal retrieval benchmark containing approximately 33,000 molecule-description pairs sourced from the ChEBI and PubChem databases. The core task is to retrieve a specific molecule from a large corpus using only its natural language description as a query. This requires a model to learn a shared semantic space bridging textual descriptions and molecular graph structures. Unlike versions used for role classification, this dataset is specifically designed to test a model's ability to understand fine-grained, descriptive text for direct molecule retrieval.

**Language-Plus-Molecule-24 Dataset:** The Language-Plus-Molecule-24 (LPM-24) dataset is a large-scale, multimodal dataset designed for text-molecule tasks, particularly cross-modal retrieval and molecule captioning (Edwards et al., 2024). It was created by mining US patent documents to extract pairs of molecule structures and their corresponding textual descriptions. LPM-24 is significantly larger and more diverse in its language complexity than previous benchmarks like ChEBI-20. The descriptions are often highly technical, focusing on the molecule's synthesis, properties, or application. This makes it a challenging benchmark for training and evaluating models on their ability to comprehend and align complex, domain-specific language with intricate molecular structures.

**USPTO-50K Dataset:** The USPTO-50K (Schneider et al., 2016) dataset for retrosynthesis is a benchmark derived from the standard USPTO-50k reaction prediction dataset. The goal is to predict a set of plausible reactants (precursors) that could synthesize a given product (target molecule). Models are trained to learn the logic of chemical disconnection and identify the key bonds to break in the target molecule. This dataset is fundamental for training and evaluating machine learning models, especially graph-based and transformer models, on their ability to perform single-step retrosynthesis, which is a critical component of computer-aided synthesis planning.

**USPTO-Mixed Dataset:** USPTO-Mixed (Jin et al., 2017) data is also derived from USPTO patent. Its corresponding task is to perform forward reaction prediction. It mixed the reactants and reagent

to make the task even more challenging for downstream model to identify key reaction and generate correct products.

## B  EXPERIEMENT DETAILS

This section provides detailed experimental setup for reproducability. The anonymous code repo is here: https://anonymous.4open.science/r/molrl-1C37/README.md

**Tools and Framework:** All experiments are conducted on a 8 Nvidia A40 node. The pretrained checkpoints are loaded from Huggingface. The codes are written in PyTorch. For RL training, we use GRPO (Shao et al., 2024) from TRL (von Werra et al., 2020) package. We use vLLM (Kwon et al., 2023) for both completion generation in the forward function and round-trip consistency evaluation in the backward function. For finetuneing, we use Hugginface's trainer.

**Training hyperparameters:** For all RL training, we use 20000 samples and train for one epoch. For each sample we generate 12 completions (this non-$2^n$ number is because we use 6 GPU for training, 1 GPU for completion generation, and 1 GPU for round-trip consistency evaluation). We use a per device batch size 4, and gradient accumulation steps of 8. In total, the model witness 192 sample-completion-reward triplets in one optimization step. To sample completions we use Top-K of 40, Top-P of $\{0.4, 0.9\}$ subject to validation error, temperature of 0.9. The KL-divergence weight beta is set to either 0.04 or 0.08 depending on the validation error. For all finetuning, we use the full training set to finetune for 2 epochs, with a global batch size of 256. We use a learning rate of $2e - 5$, weight decay factor of $0.001$, max gradient norm of $0.5$. We use a warm-up ratio of 0.03 to find the best learning rate, and then the learning rate follows cosine annealing schedule. All models are finetuned via Low-Rank Adaption (LoRA) (Hu et al., 2021) with a rank of 32, alpha of 32, dropout of 0.05, and only the query and value projection matrix in the LLMs are optimized.

**Text-based tasks:** In supervised RTRL, the round-trip consistency is combined with evaluated metrics of the generated outputs. The metric needs to be single scalar, so we simply add the BLEU-2, BLEU-4, METEOR, ROUGE-1, ROUGE-2, and ROUGE-L score. When combining this with the round-trip reward, we normalize it to a value between 0 and 1 add that to the round-trip reward.

**Molecule-based tasks:** In supervised RTRL, the metric is a sum of BLEU, MACCS similarity, RDKit similarity, and Morgan similarity. Other metrics are not included mostly because they are lower-the-better metric and has not upperbound.

## C  MORE EXPERIMENTAL RESULTS

Table 7 shows model performance when RTRL is applied to different base models. All experiment are in a self-supervised manner. We can see that RTRL brings improvement to all tested base LLM, showing its generalizability. This also indicates vast hidden/coverred knowledge within pretrained LLM, and enforcing round-trip consistency can be intuitive way to utilize such knowledge and improve the model. This also shows that RTRL can potentially be applied to larger model whose ability to judge its own response is better. We leave this to future work.

Table 8 and Table 9 compare RTRL performance against base model, two synthetic training version, and entropy minimization (EM). Synthetic Output means we have a domain data $X \in \mathcal{X}$, and task is mapping it to another domain $\mathcal{Y}$, we first ask the model to generate synthetic labels $Y^*$, and perform SFT on the dataset $(X, Y^*)$. Synthetic Input means the opposite: for the same $\mathcal{X}$ to $\mathcal{Y}$ mapping task, we have a domain data $Y \in \mathcal{Y}$, we first ask the model to generate synthetic $X^*$ and then SFT on $(X^*, Y)$. These two can also be considered as self-supervised learning. For EM, we follow recent work in Agarwal et al. (2025) to use the negative entropy of the generation as the reward. This method increases model's generation confidence, which is also self-supervised. From the results, we can see that RTRL still outperforms the variants. More importantly, RTRL can consistently improve over the base model on almost every metric, while the three variants can cause degradation. On the other hand, we see that the two synthetic variants can bring overall performance improvement, which aligns with earlier finding that round-trip consistency examples can improve a model's ability (Alberti et al., 2019). Synthetic Output is underperforming, as it might strengthen the existing overfitted behavior, causing worse evaluation performance. Synthetic Input shows better results, as it is implicitly enforcing a Round-trip consistency, by training the

Table 7: RTRL applied to different base models.

| Task / Model | BLEU ↑ | Lev. ↓ | Exact Match ↑ | MACCS SIM. ↑ | RDKit SIM. ↑ | Morgan SIM. ↑ | FCD ↓ | Validity ↑ |
|---|---|---|---|---|---|---|---|---|
| *USPTO-50K Retrosynthesis* | | | | | | | | |
| Qwen-8B | 0.571 | 35.589 | 0.000 | 0.638 | 0.546 | 0.519 | 22.522 | 0.962 |
| **Qwen-8B+RTRL** | **0.581** | **34.721** | **0.000** | **0.646** | **0.556** | **0.529** | **21.560** | **0.963** |
| Mol-Instruction | 0.370 | 33.138 | 0.202 | 0.779 | 0.641 | 0.601 | 8.264 | **1.000** |
| **Mol-Instruction+RTRL** | **0.397** | **30.873** | **0.216** | **0.797** | **0.642** | **0.621** | **6.678** | 1.000 |
| ChemDFM | 0.583 | 36.762 | **0.171** | 0.796 | 0.738 | 0.650 | 22.347 | 0.988 |
| **ChemDFM+RTRL** | **0.625** | **31.996** | 0.151 | **0.805** | **0.759** | **0.671** | **21.449** | **0.992** |
| *USPTO-Mixed Reaction Prediction* | | | | | | | | |
| Qwen-8B | 0.231 | 62.644 | **0.005** | 0.368 | 0.271 | 0.251 | **21.436** | **0.942** |
| **Qwen-8B+RTRL** | **0.255** | **61.644** | 0.003 | **0.386** | **0.297** | **0.286** | 21.537 | 0.932 |
| Mol-Instruction | 0.307 | 28.725 | 0.096 | 0.578 | 0.436 | 0.385 | 4.193 | **0.999** |
| **Mol-Instruction+RTRL** | **0.338** | **24.539** | **0.148** | **0.600** | **0.442** | **0.402** | **2.381** | 0.988 |
| ChemDFM | 0.845 | 8.685 | 0.559 | 0.880 | 0.831 | 0.803 | 18.818 | **0.987** |
| **ChemDFM+RTRL** | **0.857** | **7.934** | **0.601** | **0.895** | **0.851** | **0.823** | **0.171** | 0.985 |

model to agree with its backward function, and RTRL is a systematic and organized way to perform such alignment. Comparing EM and RTRL, we see that EM comes closer to RTRL compared to the synthetic methods especially in the molecule task. However, we still see large gap between EM and RTRL on the text-based task. We suspect that this is because RTRL not only boost the confidence, but also activates more hidden knowledge to self-validate such confidence, and hence performs better.

Table 8: RTRL on molecule task compared to training with synthetic Round-trip examples.

| Task / Model | BLEU ↑ | Lev. ↓ | Exact Match ↑ | MACCS SIM. ↑ | RDKit SIM. ↑ | Morgan SIM. ↑ | FCD ↓ | Validity ↑ |
|---|---|---|---|---|---|---|---|---|
| *CHEBI-20* | | | | | | | | |
| BASE | 0.846 | 15.652 | 0.546 | 0.898 | 0.779 | 0.733 | 2.088 | 0.978 |
| Synthetic Output | 0.713 | 30.175 | 0.491 | 0.888 | 0.777 | 0.732 | 10.098 | 0.974 |
| Synthetic Input | 0.820 | 17.879 | 0.548 | 0.899 | 0.778 | 0.733 | 2.112 | 0.980 |
| EM | 0.843 | 15.990 | 0.549 | 0.895 | 0.774 | 0.733 | **2.086** | 0.978 |
| **RTRL** | **0.852** | **15.022** | **0.553** | **0.901** | **0.781** | **0.735** | 2.099 | **0.982** |

Table 9: RTRL on text task compared to training with synthetic Round-trip examples.

| Task / Model | BLEU-2 ↑ | BLEU-4 ↑ | ROUGE-1 ↑ | ROUGE-2 ↑ | ROUGE-L ↑ | METEOR ↑ |
|---|---|---|---|---|---|---|
| *CHEBI-20* | | | | | | |
| BASE | 0.286 | 0.244 | 0.406 | 0.312 | 0.378 | 0.345 |
| Synthetic Output | 0.371 | 0.320 | 0.475 | 0.371 | 0.439 | 0.418 |
| Synthetic Input | 0.379 | 0.325 | 0.478 | 0.371 | 0.442 | 0.425 |
| EM | 0.390 | 0.333 | 0.486 | 0.376 | 0.448 | 0.431 |
| **RTRL** | **0.447** | **0.380** | **0.529** | **0.406** | **0.483** | **0.481** |

In Table 10, we apply RTRL to Llasmol in the supervised learning task. We can see that the model with RTRL consistently outperform the model with only supervised learning, validating that RTRL is a general enhancer rather than a tool restricted to ChemDFM.

We can additionally apply RTRL to classification tasks. Specifically, the forward function is now the classification task, such as predicting if a molecule will have a certain property. And then the backward function is asking the model to generate a molecule that will/will not have this property. This is essentially a conditional generation task that LLM is capable of. Given this observation, we conducted experiments on RTRL's ability in classification tasks. We first use SFT on the model, and then apply RTRL to this model. The results are in Table 11.

Table 11: RTRL applied to classification tasks.

| Model | BBBP ↑ | SIDER ↑ |
|---|---|---|
| Llasmol | 0.741 | 0.701 |
| Llasmol+SFT | 0.741 | 0.689 |
| **Llasmol+RTRL** | **0.757** | **0.713** |

From the result we see, RTRL can actually improve the performance of the base chemical LLM on classification tasks. This shows a higher degree of flexibility of RTRL.

Table 10: RTRL on supervised reaction-prediction-retrosynthesis task.

| Task / Model | BLEU ↑ | Lev. ↓ | Exact Match ↑ | MACCS SIM. ↑ | RDKit SIM. ↑ | Morgan SIM. ↑ | FCD ↓ | Validity ↑ |
|---|---|---|---|---|---|---|---|---|
| *USPTO-Mixed Reaction Prediction* | | | | | | | | |
| Llasmol | 0.825 | 15.178 | 0.356 | 0.840 | 0.777 | 0.734 | 2.123 | 0.998 |
| **Llasmol+RTRL** | **0.847** | **14.448** | **0.384** | **0.857** | **0.803** | **0.759** | **1.918** | **1.000** |
| *USPTO-50K Retrosynthesis* | | | | | | | | |
| Llasmol | 0.917 | 4.968 | 0.656 | 0.915 | 0.872 | 0.851 | 0.895 | **0.996** |
| **Llasmol+RTRL** | **0.925** | **4.876** | **0.680** | **0.920** | **0.878** | **0.859** | **0.877** | 0.996 |

Table 12: RTRL transferability from retrosynthesis task to other chemical tasks.

| Model | ESOL ↓ | Lipo ↓ | BBBP ↑ | Clintox ↑ | HIV ↑ | SIDER ↑ |
|---|---|---|---|---|---|---|
| Llasmol | 1.438 | 1.09 | 74.1 | 91.8 | **96.7** | **70.7** |
| **Llasmol+RTRL** | **1.359** | **1.01** | **74.6** | **93.1** | **96.7** | **70.7** |

We can also directly apply the Llasmol model trained on the retrosynthesis-reaction-prediction round-trip task to classification and regression tasks, and evaluate the transferrability of the knowledge learned in the round-trip process. From the results in Table 12, we can see that out of the 6 targets, RTRL is at least as effective as the base model, and brings improvements on 4 targets. These tasks have a very different format from the retrosynthesis tasks. But the round-trip experience makes base LLM a stronger model with better chemical understanding; its performance on other chemistry tasks also improves.

## D    QUALITATIVE EXAMPLES

This section discuss some qualitative examples comparing base models and RTRL-improved models.

Table 13: Examples of molecule captioning outputs across different iteration in iterative RTRL.

| Molecule | Base Model Caption | Iteration 1 Caption | Iteration 2 Caption | Ground Truth |
|---|---|---|---|---|
|  CC1=CCC[C@]2(C)CC[C@H](C(C)(C)O)C[C@@H]12 | The molecule is a natural product found in Hedychium spicatum with data available. | The molecule is a eudesmane sesquiterpenoid in which the eudesmane skeleton carries a hydroxy substituent at C-11 and has a double bond between C-3 and C-4. It has a role as a volatile oil component. | The molecule is a eudesmane sesquiterpenoid in which the eudesmane skeleton carries a hydroxy substituent at C-11 and has a double bond between C-3 and C-4. It has a role as a volatile oil component. It is a eudesmane sesquiterpenoid, a member of octahydronaphthalenes and a tertiary alcohol. | The molecule is a eudesmane sesquiterpenoid in which the eudesmane skeleton carries a hydroxy substituent at C-11 and has a double bond between C-3 and C-4. It has a role as a plant metabolite and a volatile oil component. It is a eudesmane sesquiterpenoid, a member of octahydronaphthalenes and a tertiary alcohol. |
|  CCCCCCCCCC(=O)OCC(COC(=O)CCCCCCCC)OC(=O)CCCCCCCCC | The molecule is a triglyceride. | The molecule is a triglyceride that is glycerol in which all three hydroxy groups have been formally esterified with capric acid. It has a role as a Mycoplasma genitalium metabolite. It is a triglyceride, a caprate ester and a triacylglycerol 10:0. | The molecule is a triglyceride obtained by formal acylation of the three hydroxy groups of glycerol by decanoic acid. It has a role as a metabolite. It is a triglyceride and a decanoate ester. | The molecule is a triglyceride obtained by formal acylation of the three hydroxy groups of glycerol by capric (decanoic) acid. It is a triglyceride and a decanoate ester. |

**Self-supervised RTRL:** Table 14 shows examples of reaction prediction. We compare ground truth with RTRL prediction and Base Model prediction. We can see that RTRL generally finds more accurate prediction and even exact matches. Table 15 shows examples of retrosynthesis. Similarly, RTRL can generate more chemically correct examples. Meanwhile, as discussed earlier in the paper,

Table 14: Examples comparing the outputs of RTRL-ChemDFM and ChemDFM on the reaction prediction task.

| Reaction | Ground Truth | RTRL Prediction | Base Model Prediction |
|---|---|---|---|
|  |  |  |  |
| CCCCC(=O)Nc1ccc([N+](=O)[O-])cc1.CCO | CCCCC(=O)Nc1ccc(N)cc1 | CCCCC(=O)Nc1ccc(N)cc1 | CCCCc1nc2ccc([N+](=O)[O-])cc2n1CC |
|  |  |  |  |
| C1CCOC1.CC(=O)Cl.CSc1nc(=O)n(C(C)C)c(=O)[nH]1.[Na] | CSc1nc(=O)n(C(C)C)c(=O)n1C(C)=O | CC(=O)n1c(=O)nc(SC)n(C(C)C)c1=O | CC(=O)CSc1nc(=O)n(C(C)C)c(=O)[nH]1 |

Table 15: Examples comparing the outputs of RTRL-ChemDFM and ChemDFM on the retrosynthesis task.

| Product | Ground Truth | RTRL Prediction | Base Model Prediction |
|---|---|---|---|
|  |  |  |  |
| CC(=O)OCc1nc2cc3c(cc2c(=O)[nH]1)CCC3 | CC(=O)[O-].O=c1[nH]c(CCl)nc2cc3c(cc12)CCC3 | O=c1[nH]c(CCl)nc2cc3c(cc12)CCC3.CC(=O)O | COC(=O)c1cc2c(cc1N)CCC2.CCOC(=O)C#N.CC(=O)OC(C)=O |
|  |  |  |  |
| O=C(O)C(CC1CCCC1)n1nccc1=O | COC(=O)C(CC1CCCC1)n1nccc1=O | COC(=O)C(CC1CCCC1)n1ncccc1=O.[OH-].[Na+] | O=c1cccn[nH]1.C=C(CC1CCCC1)C(=O)OCC |

we also see RTRL prefers to generate a complete reaction, like the example in the second row, where the main reactant is captured fully.

**Iterative RTRL:** Table 13 shows generated examples when we use iterative RL to improve a model's molecule captioning ability. The two examples both show that the first iteration will teach the model to generate more specified and informative example, while the second iteration will further refine the model from the first iteration. However, we also notice interesting difference between the two example. In the first example, iteration 2 caption adds in missing details to iteration 1 caption. On the other hand, in the second example, iteration 2 caption removes erroneous description from iteration 1 caption. This shows that the model does not merely learn to generate longer responses especially in the iterative scenario, it can correct itself via round-trip consistency.

## E   MORE ABOUT GRPO

The RTRL framework proposed in the paper can be extended to various RL method. In this paper, we use GRPO (Shao et al., 2024) to instantiate the framework, and we introduce GRPO here for completeness.

GRPO is a RL algorithm that finetunes a policy LLM without a explicit value function. For a prompt $t$, the policy LLM generates a group of responses $G$, and the advantage for each generation in the group is computed as the normalized advantage over the average group performance:

$$\hat{A}_i = \frac{r_i - mean(r_1, ..., r_{|G|})}{std(r_1, ..., r_{|G|}) + \epsilon_{norm}} \quad \forall i \in G, \tag{12}$$

where $\epsilon_{norm}$ is a small value for numerical stability and $r_i$ is the scaler reward for each sample, and the advantages are used as signal to guide the policy update. GRPO uses a clipped objective to optimize the model,

$$\mathcal{L}(\theta) = -\frac{1}{|G|} \sum_i^{|G|} min(\frac{\pi_\theta(x_i)}{\pi_{\theta_{old}}(x_i)} \hat{A}_i, clip(\frac{\pi_\theta(x_i)}{\pi_{\theta_{old}}(x_i)}, 1-\epsilon, 1+\epsilon) \hat{A}_i) + \beta KL(\pi_\theta(x_i), \pi_{\theta_{old}}(x_i)),$$

(13)

where $x_i$ are the input sample, $\beta$ is hyperparameter controling the effect of the KL divergence term, $\pi_\theta$ is a policy/reference under the parameter $\theta$. This method belongs to the class of trust-region method, to contraint the magnitude of each upgrade. More details can be found in the original paper (Shao et al., 2024).

## F  MORE DISCUSSION ON REWARD HACKING

In practice, our format reward approach to eliminate reward hacking follows the principle in GRPO when building format reward. In GRPO, the reward contains two parts: if the answer is correct, the answer reward is 1, and otherwise 0; if the format is correct, the format reward is 1, and otherwise 0. In RTRL, the reward also consists of two parts, the round-trip log-likelihood, and the format score $\alpha F$. The key difference is that the range of the round-trip log-likelihood is not set. We know its upper bound (0), but we do not know its lower bound. To align with GRPO, the format reward is set as follows: if the format is correct, F=1; otherwise, F=0. Then, what we do is compute the round-trip log-likelihood for several steps, and get its minimum value $m$, we set $\alpha = -m$. The goal is to achieve the effect such that a format-incorrect answer, no matter how good it is in round-trip log-likelihood, cannot have a higher reward than a format-correct answer, like in GRPO. So, there is no tuning on $\alpha$; it is a hyperparameter we use to align with the principle used in GRPO. Users can adjust $\alpha$ otherwise if they want to emphasize format correctness.

In reality, while we introduced the format reward as a general technique to fight reward-hacking, we only encountered serious reward-hacking in the USPTO-mixed dataset, where the model simply copies input reactants as the product, resulting in high but meaningless round-trip consistency. And, in this case, by using the format reward, we successfully eliminated reward hacking, and the model can have both better self-consistency and performance, as shown in the experiment section. For other datasets, the model will achieve very similar results even without a format reward. Essentially, these advanced LLMs are trained with good instructions. It is not easy to hack the reward. The model will have a much higher log-likelihood when using the expected input than using the response itself as input. This is because using response as the question is against the instruction, and hence, will have low log-likelihood. Also, it is very unlikely for the forward model to generate a response that disobey the forward instruction. This explains that in a well-instructed LLM, reward-hacking will not always happen. And the format reward is sufficient in our case. Ultimately, the strong performance is our primary goal, and if the model can consistently perform on forward tasks, like what RTRL does, the reward hacking is not a primary issue.

It is certainly possible to develop more advanced techniques to reduce reward-hacking, such as a min-max framework that requires the forward generation to be dissimilar to the input, or another LLM as a judge model to detect hacking. However, we think this might be beyond the scope of the contribution of this paper on the effectiveness of round-trip consistency, and we look forward to working on this as an important future direction on more hack-prone tasks.

## G  LLM USAGE

LLM is used to polish writing in Section 1 and Section 4. All generated writing is verified and fact-checked. The related work section involves zero LLM generation.

