# OpenReview forum: "Round-trip Reinforcement Learning: Self-Consistent Training for Better Chemical LLMs"
_ICLR.cc/2026/Conference — Submitted to ICLR 2026_

### Official Review · Reviewer_uzvi · 2025-10-20

**Soundness:** 3
**Presentation:** 3
**Contribution:** 3
**Rating:** 4
**Confidence:** 5

**Summary:**

The paper introduces **Round-Trip Reinforcement Learning (RTRL)**, a training framework designed to explicitly improve **round-trip consistency (RTC)** in chemical LLMs.
The authors observe that existing chemical LLMs perform well in one direction (e.g., for reaction prediction or retrosynthesis) but fail to reproduce the original input when performing the inverse mapping (e.g., reaction prediction and then retrosynthesis to get the raw input).

To address this, the authors propose to treat RTC not just as an evaluation metric but as a trainable RL objective. In RTRL, the model generates a forward output and is rewarded based on how well a backward model can reconstruct the original input. They employ GRPO for stable RL optimization and further introduce an iterative self-improvement loop where forward and backward tasks train each other alternately. Experiments on round-trip chemical tasks (e.g., **reaction prediction/retrosynthesis and molecule captioning/generation**) show up to ~52% gain in round-trip consistency and ~55% improvement in primary task metrics.

**Strengths:**

This paper focuses on some of the most important tasks in molecule-text modeling and achieves obvious performance improvements through the following advantages:
- **Novel training objective with domain grounding:** The paper innovatively transforms round-trip consistency from a diagnostic metric into a direct optimization target. This aligns well with the inherently bidirectional nature of many chemical phenomena.
- **Elegant self-improvement design:** The iterative RTRL loop enables mutual reinforcement between forward and backward mappings, leading to better bidirectional performance without relying solely on paired data.
- **Good writing quality and clarity:** The paper is logically structured, well written, and progresses in a clear, step-by-step manner—making complex RL-based ideas accessible to a broader audience.

**Weaknesses:**

Below are my concerns about this paper. My major concerns are about weaknesses 3 and 4: about the improvement of performances on forward and backward tasks (instead of the RTC ones). I would be happy to raise my score if the author can provide satisfactory response.
- **Framework figure complexity:** The framework figure in the paper (Figure 1) is visually cluttered and difficult to parse. Its layout and dense notation make it hard for readers to quickly understand the overall workflow and the interaction between the forward model, backward model, and RL loop.
- **Potential for reward hacking:** As with any RL setup, the model might exploit trivial output patterns that are easy to invert but chemically meaningless. For example, what if the forward LLM includes the molecular SMILES in some corner when generating its caption? This would make the backward task super easy.
- **Mixed experimental outcomes:** While consistency metrics improve significantly, the actual results do not hold across all tasks (e.g., some relatively low or even negative improvements in Tables 2 and 3) and are not particularly strong compared to baselines. This may limit enthusiasm for immediate practical adoption.
- **Weak baselines:** The baseline models included in this research are either general LLMs (GPT) or relatively old—the latest model is ChemDFM, which was proposed in Jan 2024. Comparing with stronger or more influential baseline models (e.g., Mol-LLM, InstructMol-GS, MolecularGPT, ReactXT) could better strengthen the findings in this paper.

[1] Mol-LLM: Multimodal generalist molecular LLM with improved graph utilization
[2] MolecularGPT: Open large language model for few-shot molecular property prediction
[3] InstructMol: Multi-modal integration for building a versatile and reliable molecular assistant in drug discovery, COLING 2025
[4] ReactXT: Understanding Molecular “Reaction-ship” via Reaction-Contextualized Molecule-Text Pretraining, ACL 2024

**Questions:**

I'm interested in more discussion regarding the reward hacking phenomenon. As mentioned in the Weaknesses section, I don't think a single format reward calculated by RDKit can eliminate this problem in such an iterative self-improving setting.

---

> ### Author Response · Authors · 2025-11-20
> **Author response (1/2)**
>
> Dear Reviwer uzvi,
>
> We sincerely appreciate your thorough evaluation of our work. There are many key points that we found particularly fundamental in both refining and presenting our paper. Here we provide responses to your concerns.
>
> > W1: Figure too dense.
>
> Thank you for pinpointing the uncertainty in the main figure. Based on your suggestion, we revised the main pipeline. We factor out the main components (forward, backward, and RL), so their connection and pipeline are easier to capture. The new figure can be found in the current revision. We will update the revision again to contain the rest of the content after our discussion.
>
> > W2, Q1: In the round-trip scenario, how to avoid reward hacking?
>
> We would like to further elaborate on how the format reward functions. Our approach actually **follows the principle in GRPO**. In GRPO, the reward contains two parts: if the answer is correct, the answer reward is 1, and otherwise 0; if the format is correct, the format reward is 1, and otherwise 0. In RTRL, the reward also consists of two parts, the round-trip log-likelihood, and the format score $\alpha F$. The key difference is that the range of the round-trip log-likelihood is not set. We know its upper bound (0), but we do not know its lower bound. To align with GRPO, the format reward is set as follows: if the format is correct, F=1; otherwise, F=0. Then, what we do is compute the round-trip log-likelihood for several steps, and get its minimum value $m$, we set $\alpha=-m$. The goal is to achieve the effect such that **a format-incorrect answer, no matter how good it is in round-trip log-likelihood, cannot have a higher reward than a format-correct answer**, like in GRPO. So, there is **no tuning** on $\alpha$; it is a hyperparameter we use to align with the principle used in GRPO. We make $\alpha$ a tunable parameter so that our framework is general if a user wants to emphasize format correctness.
>
> Then, we would like to discuss the reality in our development. While we introduced the format reward as a general technique to fight reward-hacking, *we only encountered serious reward-hacking in the USPTO-mixed dataset*, where the model simply copies input reactants as the product, resulting in high but meaningless round-trip consistency. And, in this case, by using the format reward, **we successfully eliminated reward hacking**, and the model can have both better self-consistency and performance, as shown in the experiment section. For other datasets, the model will achieve very similar results even without a format reward. Essentially, these advanced LLMs are trained with good instructions. It is not easy to hack the reward.  For example, we did the following case study, *like the cases you suggested*. We compute the log-likelihood of the response, given the expected input and the response itself, on the molecule-caption tasks.
>
> |  | Molecule-Caption | Caption-Molecule |
> |---|---|---|
> | Molecule | **-0.231+-0.120** | -0.285+-0.069 |
> | Caption | -0.457+-0.103 | **-0.182+-0.075** |
>
> Columns represent the input, and row represents the instruction. From the results, we see that when presented with the expected input, the model actually has a much higher log-likelihood than using the response itself as input. This is because **using response as the question is against the instruction**, and hence, will have low log-likelihood. Also, it is very unlikely for the forward model to generate a response that disobey the forward instruction. This explains that in a well-instructed LLM, reward-hacking will not always happen. And the format reward is sufficient in our case. Ultimately, **the strong performance is our primary goal**, and if the model can consistently perform on forward tasks, like what RTRL does in the paper, the reward hacking is not a primary issue. And the good performance, conversely, provides validation that our approach is effective in mitigating reward hacking.
>
> Please continue to the second part of our response.

---

> ### Author Response · Authors · 2025-11-20
> **Author Response (2/2)**
>
> Continued discussion on reward-hacking
>
> > W2, Q1: In the round-trip scenario, how to avoid reward hacking?
>
> From a broader perspective, reward-hacking prevention itself is a difficult and interesting subject. It is more of a task-specific technique. There is no method in the literature that can provide a guarantee and proof that reward-hacking will not happen. It usually emerges from empirical evidence and intuition. For example, format reward adds a hard constraint, adjusting the KL-divergence constant $\beta$ regularizes the model to not deviate from normal behavior in PPO, and iterative RTRL introduces dynamics into the system, so it is difficult to hack. These examples show that fighting reward-hack is a fluid process without a ground-truth answer. Hence, people adopt different approaches depending on scenarios. In areas like code generation, people rely on tag extraction (finding Python/Java markdown blocks) and a compiler to prevent unexpected behavior that causes reward hacking. In our case, because RDKit can serve as a deterministic tool to validate a completion in the chemical domain, we use it to solve the reward-hacking problem, and it worked very well.
>
> It is certainly possible to develop more advanced techniques to reduce reward-hacking, such as a min-max framework that requires the forward generation to be dissimilar to the input, or another LLM as a judge model to detect hacking. However, we think this might be beyond the scope of the contribution of this paper on the effectiveness of round-trip consistency, and we look forward to working on this as an important future direction on more hack-prone tasks.
>
>
> > W4: Weak baselines.
>
> We would like to first clarify to you that we picked ChemDFM not because we want to avoid comparison with a more competitive model, but rather, none of the existing chemical LLM shows unanimous superiority on all tasks. You and one of the other reviewers suggested several critical baselines. *We believe they are very relevant and inspiring, and we will discuss them in the paper*. Meanwhile, after careful examination of their efficacy and availability, we will use Llasmol as a strong baseline to conduct further experiments (note that ChemDFM outperforms it on caption-molecule tasks). It achieves similar performance to Mol-LLM in your suggested works, and we will certainly include a comparison to Mol-LLM when its model becomes available.
>
> Here we show RTRL's performance when applied to Llasmol on retrosynthesis and reaction prediction tasks (base Llasmol outperforms base ChemDFM on these tasks):
>
> |  | BLEU | EM | Lev. | Val | MACCS Sim. | RDkit Sim. | Morgan Sim. | FCD |
> |---|---|---|---|---|---|---|---|---|
> | Reaction Prediction Llasmol | 0.825 | 0.356 | 15.178 | 0.998 | 0.840 | 0.777 | 0.734 | 2.123 |
> | Reaction Prediction Llasmol+RTRL | **0.847** | **0.384** | **14.448** | **1.0** | **0.857** | **0.803** | **0.759** | **1.918** |
> | Retrosynthesis Llasmol | 0.917 | 0.656 | 4.968 | 0.996 | 0.915 | 0.872 | 0.851 | 0.895 |
> | Retrosynthesis Llasmol+RTRL | **0.925** | **0.680** | **4.876** | **0.996** | **0.920** | **0.878** | **0.859** | **0.877** |
>
> From the results, we can see that even when applied to a new base model, RTRL can still bring considerable and consistent improvement. Validating the generalizability of RTRL.
>
> > W3: Mixed experimental outcomes
>
> We agree with you that RTRL did not achieve the best performance on every metric. For example, you mentioned in Tables 2 and 3 that RTRL causes negative improvement on some metrics. In Table 2, while RTRL slightly falls short on FCD (20.723 vs 19.987), it **significantly improves exact match** (0.821 vs 0.547). We believe RTRL's effectiveness is consistent and significant on most metrics. In Table 3, we can see that RTRL actually outperforms its base model ChemDFM on almost every metric. On metrics where RTRL is outperformed by base Mol-instruction, we invite you to **Appendix Table 7**, where we also applied RTRL to Mol-instruction, and we can see RTRL is still effective when the base model changes. Also, in our response above, we additionally provide RTRL performance when applied to Llasmol. RTRL is still highly effective, showing good generalizability and a wide range of applicable scenarios of RTRL.
>
>
> Thank you again for providing such a detailed and insightful response to us. We look forward to your reply. And we will include all of our discussion in the revised version.

---

> ### Author Response · Authors · 2025-11-28
> **Looking forward to your reply**
>
> Dear Reviewer uzvi,
>
> As we have roughly 4 days left in the author-reviewer discussion period, we just wanted to check and see if our response is satisfactory to address your concerns. Mainly, we provided responses on when reward hacking in RTRL happens, how we solve it, why the current approach is sufficient, and what we plan to improve. We further provided results on stronger baselines, and showed that RTRL is a general enhancer. We also greatly appreciate your detailed look, and we have improved the figure in the current submission. Again, we would love to engage in more discussion with you about our work, and feel free to let us know any further questions and concerns.

---

### Official Review · Reviewer_ifiU · 2025-10-30

**Soundness:** 3
**Presentation:** 3
**Contribution:** 3
**Rating:** 6
**Confidence:** 3

**Summary:**

This paper addresses the lack of "Round-Trip Consistency" (RTC) in Large Language Models (LLMs) in chemistry when handling bidirectional tasks (such as reaction prediction and retrosynthesis, molecular description and generation). A novel training framework called "Round-Trip Reinforcement Learning" (RTRL) is proposed. The authors observe that existing models may successfully map from A to B, but fail to accurately reconstruct A from B, indicating that the model learns a unidirectional, shallow memory rather than a flexible understanding of chemical principles.

The core idea of ​​RTRL is to treat RTC as a trainable objective, using the success or failure of the ring transition as a reward signal, and optimizing the model through reinforcement learning (specifically the GRPO algorithm). The authors design an efficient reward function, using the conditional likelihood $p(x|y)$ of the original input $x$ at the intermediate output $y$ as the surrogate reward, avoiding expensive double generation and complex similarity metrics.

Furthermore, the framework supports Iterative RTRL, enabling forward and backward models (both powered by the same LLM) to mutually reinforce each other. This approach demonstrates effectiveness across various scenarios, including self-supervised (using only unlabeled data), supervised, and synthetic data. Experimental results show that RTRL significantly improves the model's RTC (up to 52%) and main task performance (up to 55%).

**Strengths:**

1. Conceptual Novelty. The most significant contribution of this paper lies in its novel perspective: transforming "Round-Trip Consistency" from a traditional evaluation metric into a directly optimizable training objective.

2. Efficiency. By using conditional log-likelihood $\log p_{\phi}(x|y, t_g)$ as a reward, this method (1) avoids the complete backward generation step, greatly improving computational efficiency; (2) by using LLM itself as a "referee", it bypasses the difficulty of designing effective text similarity measures (such as BLEU) in the field of chemistry (such as SMILES).

3. Comprehensive Evaluation. The authors conducted a series of well-designed experiments that strongly support their core arguments. Experiments cover two core bidirectional tasks: reaction prediction (retrosynthesis) and molecular description (text-based molecule generation).

**Weaknesses:**

1. Need more discussion on Reward Hacking. The authors mention the possibility of "reward hacking," where a model might learn to copy the input ($A \rightarrow A$) to easily obtain a high reward of $p(x|y)$. Authors propose a solution by introducing an optional "format score" $F(y)$ (Eq. 10). This solution is somewhat ad-hoc; the Authors do not discuss the sensitivity of the hyperparameter $\alpha$ in detail, nor do they explore whether such a simple format check is sufficient to prevent more sophisticated "cheating" behavior.

2. Stability and Scalability Concerns. This method relies on Reinforcement Learning (GRPO), and the training stability of applying RL to LLM is a well-known challenge. The authors mention that they had to freeze the parameters of the reward model $\phi$ to stabilize training, implying the sensitivity of the training process. Experiments were conducted on an 8B model. While the authors speculate that RTRL can be applied to larger models, this has not been confirmed. Furthermore, RTRL requires generating multiple completions per sample, which could incur high computational costs when scaling to larger models.

**Questions:**

Regarding the robustness of the reward mechanism and "reward hacking":

1. How sensitive is this method to the hyperparameter $\alpha$? Does the setting of $\alpha$ require fine-grained manual tuning for different tasks?

2. (Important) How can this simple format check guarantee against more "smart" cheating? For example, the model could learn to generate an intermediate output $y$ that is chemically meaningless or very mediocre, but structurally easy to reverse engineer. This would still maximize your surrogate reward $\log p_{\phi}(x|y)$, but completely defeat the purpose of improving task performance. How do you prove that RTRL incentivizes genuine task improvement, and not just an improvement in "reversibility"?

---

> ### Author Response · Authors · 2025-11-20
> **Author response (1/2)**
>
> Dear Reviewer ifiU:
>
> We would like to first thank you for your critical comments and positive feedback on our paper. They are invaluable resources for us to keep improving our work. Here, we extend our discussion with the following responses to your concerns:
>
> > W1, Q2: You raised several concerns on the reward hacking phenomenon in RTRL. Mainly, the forward function can learn trivial patterns with high round-trip consistency, and the hyperparameter $\alpha$ seems arbitrary.
>
> We would like to further elaborate on how the format reward functions. You mentioned that our approach is somewhat ad hoc. In fact, our approach attempts to **follow the principle in GRPO**. In GRPO, the reward contains two parts: if the answer is correct, the answer reward is 1, and otherwise 0; if the format is correct, the format reward is 1, and otherwise 0. In RTRL, the reward also consists of two parts, the round-trip log-likelihood, and the format score $\alpha F$. The key difference is that the range of the round-trip log-likelihood is not set. We know its upper bound (0), but we do not know its lower bound. To align with GRPO, the format reward is set as follows: if the format is correct, F=1; otherwise, F=0. Then, what we do is compute the round-trip log-likelihood for several steps, and get its minimum value $m$, we set $\alpha=-m$. *The goal is to achieve the effect such that a format-incorrect answer, no matter how good it is in round-trip log-likelihood, cannot have a higher reward than a format-correct answer, like in GRPO*. So, there is no tuning on $\alpha$; it is a hyperparameter we use to align with the principle used in GRPO. We make $\alpha$ a tunable parameter so that our framework is general if a user wants to emphasize format correctness.
>
> Then, we would like to discuss the **reality** in our development. While we introduced the format reward as a general technique to fight reward-hacking, we **only** encountered serious reward-hacking in the USPTO-mixed dataset, where the model simply copies input reactants as the product, resulting in high but meaningless round-trip consistency. And, in this case, *by using the format reward, we successfully eliminated reward hacking*, and the model can have both better self-consistency and performance, as shown in the experiment section. For other datasets, the model will achieve very similar results even without a format reward. Essentially, these advanced LLMs are trained with good instructions. It is not easy to hack the reward. For example, we did the following case study. We compute the log-likelihood of the response, given the expected input and the response itself, on the molecule-caption tasks.
>
> |  | Molecule-Caption | Caption-Molecule |
> |---|---|---|
> | Molecule | **-0.231+-0.120** | -0.285+-0.069 |
> | Caption | -0.457+-0.103 | **-0.182+-0.075** |
>
> Columns represent the input, and row represents the instruction. From the results, we see that when presented with the expected input, the model actually has a much higher log-likelihood than using the response itself as input. This is because using the response itself as the question is against the instruction, and hence, will have low log-likelihood. Also, it is very unlikely for the forward model to generate a response that disobey the forward instruction. This explains that in a well-instructed LLM, **reward-hacking will not always happen**. And the format reward is sufficient in our case. Ultimately, **the strong performance is our primary goal**, and if the model can consistently perform on forward tasks, like what RTRL does in the paper, the reward hacking is not a primary issue. And the good performance, conversely, provides validation that our approach is effective in mitigating reward hacking.
>
> Please continue to the second part of the response.

---

> ### Author Response · Authors · 2025-11-20
> **Author response (2/2)**
>
> Continued discussion reward hacking
>
> > W1, Q2: You raised several concerns on the reward hacking phenomenon in RTRL. Mainly, the forward function can learn trivial patterns with high round-trip consistency, and the hyperparameter $\alpha$ seems arbitrary.
>
> From a broader perspective, reward-hacking prevention itself is a difficult and interesting subject. There is no method in the literature that can provide a guarantee and proof that reward-hacking will not happen. It usually emerges from empirical evidence and intuition. For example, format reward adds a hard constraint, adjusting the KL-divergence constant $\beta$ regularizes the model to not deviate from normal behavior in PPO, and iterative RTRL introduces dynamics into the system, so it is difficult to hack. These examples show that fighting reward-hack is a fluid process without a ground-truth answer. Hence, people adopt different approaches depending on scenarios. In areas like code generation, people rely on tag extraction (finding Python/Java markdown blocks) and a compiler to prevent unexpected behavior that causes reward hacking. In our case, because RDKit can serve as a deterministic tool to validate a completion in the chemical domain, we use it to solve the reward-hacking problem, and it worked very well.
>
> It is certainly possible to develop more advanced techniques to reduce reward-hacking, such as a min-max framework that requires the forward generation to be dissimilar to the input, or another LLM as a judge model to detect hacking. However, we think this might be beyond the scope of the contribution of this paper on the effectiveness of round-trip consistency, and we look forward to working on this as an important future direction on more hack-prone tasks.
>
> > W2: Stability in optimizing RTRL.
>
> We agree with you that optimizing the model with RL is less stable. And many works have been exploring this drawback of RL for LLM. We freeze part of the system, and freezing part of the system is actually the standard practice [1] for improved efficiency and achieving good effectiveness. On the other hand, compared to rewards like validity and exact match, which are sparse (0/1), the reward in RTRL is **continuous/dense (log-likelihood)**. And it is a well-known phenomenon that continuous reward can lead to more stable training. Hence, compared to the traditional RL method that uses manually-designed sparse rewards, the log-likelihood reward can, in fact, be more stable and result in better results, as we see in the experiment section.
>
> > W2: Scalability is not concretely demonstrated in the paper.
>
> We also would love to extend the method to large-scale models. However, almost all chemical LLM falls in the 7-8B range, and our method is built on chemical LLMs; we do not have the resources to pretrain a chemical LLM. We will apply RTRL to a larger model to examine its scalability as more advanced models become available.
>
> We are very grateful for your valuable time spent on helping us improve our paper, and we hope our responses address your concerns. And, we would love to extend our discussion further. We will include all of our discussion and new results in our revision.
>
> [1] Fu, Wei, et al. "AReaL: A Large-Scale Asynchronous Reinforcement Learning System for Language Reasoning." arXiv preprint arXiv:2505.24298 (2025).

---

> ### Author Response · Authors · 2025-11-28
> **Looking forward to your reply**
>
> Dear Reviewer ifiU,
>
> Once again, thank you for your positive feedback, and we just want to check in and see if our follow-up experiments addressed your concerns? Mainly, we provided detailed analysis about reward hacking phenomenon in RTRL, and why the current approach is a valid approach for RTRL. We also provided further discussion on stability and scalabilty of RTRL. We strive to learn from all reviewers and improve our work, so we will greatly appreciate it if you could kindly let us know whether our responseses can address the weaknesses you mentioned. Looking forward to your reply.

---

### Official Review · Reviewer_K6qN · 2025-11-01

**Soundness:** 2
**Presentation:** 3
**Contribution:** 1
**Rating:** 2
**Confidence:** 5

**Summary:**

The paper introduces Round-Trip Reinforcement Learning (RTRL), a framework designed to address the critical issue of round-trip consistency (RTC) failure in chemical Large Language Models (LLMs). This inconsistency suggests models rely on shallow, unidirectional memorization rather than robust, bidirectional mastery of chemical principles. The experimental results show that the proposed RTRL method can help improve the round-trip consistency and also improve the performance of base model. However, the effectiveness of the proposed method is only validated on paired round-trip tasks and is not validated on general chemistry tasks. In addition, the performance comparison lacks significant SOTA on some tasks, making the conclusion to be less well-supported.

**Strengths:**

1. Fosters deeper bidirectional coherence

By treating consistency as a direct, trainable objective, RTRL encourages the model to move beyond shallow, unidirectional memorization.

2. Efficient utilization of unlabeled data

RTRL employs a self-supervised objective that is highly data-efficient. Training only requires access to inputs from a single domain (X), such as a list of molecules, without needing corresponding paired, labeled data.

3. Measurable performance and consistency gains

RTRL provides measurable gains over strong baselines, significantly boosting model performance and consistency.

**Weaknesses:**

1. **Limited generalizability and validation scope**

The approach is demonstrated exclusively on bidirectional generative tasks (such as molecule captioning ⇌ generation and reaction prediction ⇌ retrosynthesis). The core conclusion that round-trip consistency is critical is based on these specific tasks, and the framework is not validated on general tasks that are not round-trip tasks. Furthermore, RTRL currently does not work with classification or regression tasks.

2. **Performance comparison not comprehensive enough**

While RTRL significantly improves the base model it is applied to (ChemDFM), the resulting absolute performance figures on certain tasks may not represent the highest State-of-Art (SOTA) benchmarks. For example, on the USPTO-50K Retrosynthesis task in the self-supervised setting, the Exact Match accuracy achieved was 0.151, the SOTA in the paper achieved 0.202. But the Llasmol [1] has the exact match accuracy 0.329, significantly higher than the SOTA baseline in the paper and the proposed method.

[1] Yu, Botao, et al. "Llasmol: Advancing large language models for chemistry with a large-scale, comprehensive, high-quality instruction tuning dataset." arXiv preprint arXiv:2402.09391 (2024).

**Questions:**

1.  Can the proposed method be helpful for other non-round-trip-paired tasks or other chemistry tasks?

2.  Augmenting the round-trip paired tasks with the same dataset has been explored well in chemistry such as [1]. Can the proposed method be compared with those baselines that are simply trained with multiple combined datasets like Llasmol? It will also be great to include some chemistry-specific method like [1].


[1] Tetko, Igor V., et al. "State-of-the-art augmented NLP transformer models for direct and single-step retrosynthesis." Nature communications 11.1 (2020): 5575.

---

> ### Author Response · Authors · 2025-11-20
> **Author Response (1/2)**
>
> Dear Reviewer K6qN,
>
> We'd like to first thank you for your time devoted to providing critical feedback and helping us improve our work. Here, we provide responses and new experimental results as additional reference for you to evaluate our paper.
>
> > W2: The paper uses ChemDFM as the base model, which does not seem to be the SOTA model. Does RTRL still work as the base model gets stronger and stronger?
>
> We would like to first clarify to you that we pick ChemDFM not because we want to avoid comparison with a more competitive model, but rather *none of the existing chemical LLM shows unanimous superiority on all tasks*. For example, the work you mentioned, **Llasmol** [1], is outperformed by ChemDFM on molecule-text related tasks. We pick ChemDFM because it uses a newer base LLM (LLaMA-3), and it has good overall performance, community support, and it is also a model already trained with a **combined dataset** as you mentioned (e.g., both reaction prediction and retrysynthesis like Llasmol).
>
> At the same time, we agree with you that this is a valid and important concern. *Showing that RTRL works across different base models is critical for RTRL to be a general approach to augment LLMs*. We are aware of this problem, and in **Appendix C Table 7** of the original submission, we already included RTRL performance on other base models, including vanilla Qwen, chemical LLM Mol-Instruction, and RTRL shows consistent improvement.
>
> We agree that Llasmol is indeed a fundamental chemical LLM to evaluate against. Hence, to take our evaluation one step further, in this rebuttal, *we will apply RTRL to Llasmol as you suggested*, and show that RTRL is a generic enhancer rather than an exclusive tool for ChemDFM. First, we show RTRL's performance when applied to Llasmol on retrosynthesis and reaction prediction tasks (base Llasmol outperforms base ChemDFM on these tasks):
>
> |  | BLEU | EM | Lev. | Val | MACCS Sim. | RDkit Sim. | Morgan Sim. | FCD |
> |---|---|---|---|---|---|---|---|---|
> | Reaction Prediction Llasmol | 0.825 | 0.356 | 15.178 | 0.998 | 0.840 | 0.777 | 0.734 | 2.123 |
> | Reaction Prediction Llasmol+RTRL | **0.847** | **0.384** | **14.448** | **1.0** | **0.857** | **0.803** | **0.759** | **1.918** |
> | Retrosynthesis Llasmol | 0.917 | 0.656 | 4.968 | 0.996 | 0.915 | 0.872 | 0.851 | 0.895 |
> | Retrosynthesis Llasmol+RTRL | **0.925** | **0.680** | **4.876** | **0.996** | **0.920** | **0.878** | **0.859** | **0.877** |
>
> From the results, we can see that even when applied to a new base model, RTRL can still bring considerable and consistent improvement. Validating the generalizability of RTRL
>
> > W1: The generalizability and validation scope of RTRL. The question is that, in the paper, RTRL is only evaluated on a round-trip task; does RTRL work on other tasks?
>
> This is a great and critical question. It demands RTRL to be versatile across different tasks and scenarios. We address this concern from three scopes:
>
> **From the training scope**, can we apply RTRL to non-round-trip task? For example, classification. While we acknowledged that classification is difficult for RTRL to do in our original submission, we found that in our follow-up study, **classification still falls under the RTRL framework**. Specifically, the forward function is now the classification task, such as predicting if a molecule will have a certain property. And then the backward function is asking the model to generate a molecule that will/will not have this property. This is essentially a conditional generation task that LLM is capable of. Given this observation, we conducted experiments on RTRL's ability in classification tasks. We first use SFT on the model, and then apply RTRL to this model.
>
> |  | BBBP | SIDER |
> |---|---|---|
> | Llasmol | 0.741 | 0.701 |
> | Llasmol+SFT | 0.741 | 0.689 |
> | Llasmol+RTRL | **0.757** | **0.713** |
>
> From the result we see, RTRL can actually improve the performance of the base chemical LLM on **classification tasks**. This shows a higher degree of flexibility of RTRL. And we also see that training with *simple round-trip examples using SFT for the backward function is not sufficient to improve the forward ability*.
>
> Please continue to the second part of the response.

---

> ### Author Response · Authors · 2025-11-20
> **Author Response (2/2)**
>
> Continued discussion on the generalizability of RTRL
>
> > W1: The generalizability and validation scope of RTRL. The question is that, in the paper, RTRL is only evaluated on a round-trip task; does RTRL work on other tasks?
>
> Second, **from the inference scope**, we trained RTRL on these so-called round-trip tasks. Does the model only improve on the round-trip tasks, or can the knowledge transfer to other tasks? For this, we directly apply the Llasmol model trained on the retrosynthesis-reaction-prediction round-trip task to classification and regression tasks.
>
> |  | ESOL | Lipo | BBBP | Clintox | HIV | SIDER |
> |---|---|---|---|---|---|---|
> | Llasmol | 1.438 | 1.09 | 74.1 | 91.8 | 96.7 | 70.7 |
> | Llasmol+RTRL | **1.359** | **1.01** | **74.6** | **93.1** | 96.7 | 70.7 |
>
> From the results, we can see that out of the 6 targets, RTRL is at least as effective as the base model, and brings improvements on 4 targets. These tasks have a **very different format** from the retrosynthesis tasks. But the round-trip experience makes base LLM a **stronger model with better chemical understanding**; its performance on **other chemistry tasks** also improves.
>
>
> Third, **from the task scope**, an interesting question is what round-trip tasks are? For example, in the Llasmol paper, almost all of the evaluated tasks are round-trip, IUPAC-molecule, caption-molecule, and reactants-product. As mentioned above, classification can also be interpreted as a round-trip.
>
> The definition of round-trip tasks can be very broad. If we consider an arbitrary question Q, it can be considered as a mapping from X to Y. There is no constraint on what X and Y should look like, so the backward function is just the inverse mapping. With this mindset, we can view any question that takes an input and expects an output as an applicable round-trip forward function. And the unique versatility of RTRL, as you also mentioned in the *strength section of your review*, is that **collecting answers is not necessary**, and one only needs to come up with an inverse question to train the model. *We believe that instead of constraining the model, RTRL actually liberates it by allowing it to **utilize round-trip consistency without round-trip data***.
>
>
> > Q2: Another concern you mentioned is that augmenting the model with a round-trip example has already been explored by previous works. What is RTRL's advantage over these data augmentation approaches?
>
>
> We appreciate you bringing past work into the discussion. Yes! In the original submission, we have a dedicated section 3 and several paragraphs in the introduction discussing past work related to round-trip consistency. *We are also aware that round-trip consistent data can improve the model performance*, like in the work [2] that you mentioned, and we have already compared the performance of data augmentation with RTRL in the Appedic C Table 8 and 9 in the original submission, and showed that RTRL is better at utilizing round-trip consistency. We also perform SFT with round-trip examples on the Reaction prediction and retrosynthesis task using Llasmol. Here are the results:
>
> |  | BLEU | EM | Lev. | Val | MACCS Sim. | RDkit Sim. | Morgan Sim. | FCD |
> |---|---|---|---|---|---|---|---|---|
> | Reaction Prediction Llasmol + RT examples | 0.816 | 0.339 | 16.598 | 0.998 | 0.847 | 0.743 | 0.729 | 2.539 |
> | Reaction Prediction Llasmol+RTRL | **0.847** | **0.384** | **14.448** | **1.0** | **0.857** | **0.803** | **0.759** | **1.918** |
> | Retrosynthesis Llasmol + RT examples | **0.928** | 0.663 | 4.896 | **0.998** | 0.901 | 0.856 | 0.836 | 1.064 |
> | Retrosynthesis Llasmol+RTRL | 0.925 | **0.680** | **4.876** | 0.996 | **0.920** | **0.878** | **0.859** | **0.877** |
>
> RTRL consistently outperforms plain data augmentation. We even observe performance degradation when RT augmentation is used, potentially because these samples are seen during training, and simple round-trip augmentation does not add new knowledge but causes overfitting. On the contrary, the key advantage of RTRL over plain augmentation is that it simultaneously uses exploration and verification. As the model evolves in the RL process, it can explore LLM completion that will not be explored by augmentation using a static model. It also uses the round-trip mechanism to check the generation with its inner knowledge, which verifies the soundness of the completion.
>
> Again, we sincerely appreciate your dedicated review. We hope our response can address your concern, and we look forward to any further discussion with you. We will include all these discussions and new findings in our updated submission.
>
> [1] Yu, Botao, et al. "Llasmol: Advancing large language models for chemistry with a large-scale, comprehensive, high-quality instruction tuning dataset." arXiv preprint arXiv:2402.09391 (2024).
>
> [2] Tetko, Igor V., et al. "State-of-the-art augmented NLP transformer models for direct and single-step retrosynthesis." Nature Communications 11.1 (2020): 5575.

---

> ### Author Response · Authors · 2025-11-28
> **Looking forward to your reply**
>
> Dear Reviewer K6qN,
>
> As the author-reviewer discussion is approaching an end, we would like to check in and see if our responses addressed your concerns? For your concern on the generalizability of RTRL, we show that RTRL can work on other chemical tasks that not traditionally considered as round-trip tasks (classification), we also show model trained with RTRL has stronger chemical knowledge and can transfer to other tasks. For your concern on performance comparison, we took your advise and evaluate RTRL based on Llasmol, and showed that RTRL can still improve the base model. We hope that these new evaluation and analysis can provide additional reference and perspectives to your evaluation. We would love to extend our discussion and address any further concerns you have.

---

### Author Response · Authors · 2025-12-02
**Summary of the discussion**

Dear Area Chair,

Thank you for organizing the review process of our paper. At the end of the discussion phase, we would like to provide a summary of the our discussion with the reviewer for your reference.

From the positive side:
- Reviewers highlights the comprehensive evaluation (reviewer ifiU) and consistent performance gain (reviewer K6qN), affirming that RTRL can significantly improve the base model.

- RTRL proposes a novel and trainable objective (reviewer ifiU, uzvi) with domain grounding (reviewer uzvi), allowing RTRL to effectively train in the chemical domain.

- RTRL is efficient both in terms of data (reviewer K6qN, uzvi) and computation (reviewer ifiU), making RTRL a easy to adopt solution, even when we only have unlabelled data.

The reviewers also raises several concerns about RTRL. Two main conerns are:
- Generalizability: reviewer K6qN questioned that RTRL might only work for traditionally round-trip tasks, and is not general. In our response, we provided new results on **RTRL's transferrability across different tasks types (classification and regression)**, and its ability when directly applied to classfication tasks, showing that RTRL is indeed versatile.

- Reward hacking: reviewer ifiU and uzvi was curious about the details on how the format reward is designed and how it reduce reward hacking. We provided comprehensive discussion from multiple perspectives and illustrates that we can successfully minimize the impact of reward hacking and achieve good performance using format reward.

We provided 1-to-1 responses to reviewer's questions in our individual comments. We also updated the submission pdf to include the new experimental results and a clearer version of the pipeline illustration figure. While we didn't have time to engage in more rounds of discussion with the reviewers, we greatly appreciate both you and the reviewers for spending time on providing invaluable feedbacks, we will learn from all of your comments and improve our work comprehensively in our future revisions.

Sincerely,
Authors

---

### Meta-Review · Area_Chair_ymHz · 2026-01-07

**Summary:**

This submission addresses the issue of round-trip consistency in Large Language Models (LLMs) used for computational chemistry, highlighting that while these models can perform tasks like reaction prediction, they often fail to accurately reconstruct structures. The authors introduce Round-Trip Reinforcement Learning (RTRL), a framework that uses the success of round-trip transformations as a reward signal to improve model consistency. They further propose an iterative approach where forward and reverse mappings alternate in a self-improvement loop, making the process highly data-efficient. Experimental results demonstrate that RTRL significantly enhances both performance and consistency, offering a new direction for building more robust and reliable foundation models.

**Reviewer Concerns:**

Generally, the reviewers's concerns can be summarized as follows,

1) Limited Generalizability and Validation Scope: The proposed approach is primarily demonstrated on bidirectional generative tasks, with no validation on other general tasks such as classification or regression, limiting its broader applicability. Additionally, the framework's conclusions on round-trip consistency may not extend to non-round-trip tasks.

2) Performance Comparison and Weak Baselines: While RTRL improves the base model (ChemDFM), the performance on certain tasks does not match the highest state-of-the-art benchmarks. Moreover, comparisons are made with relatively weak or outdated baseline models, such as ChemDFM, and would benefit from comparisons with more influential models like Mol-LLM or ReactXT.

3) Stability, Scalability, and Reward Hacking Concerns: The method faces challenges related to training stability when applying reinforcement learning (RL) to LLMs, with the authors noting the need to freeze parameters to stabilize training. Additionally, potential "reward hacking" issues could arise, where the model exploits trivial patterns for high rewards, raising concerns about the robustness of the reward mechanism.

**Reviewer Scores:**

Three of four reviewers submitted their ratings, and the rating scores are respectively 2, 6, 4. Regarding the reviewers' concerns, the authors provided substantial effort in the rebuttal. As AC, it is expected the reviewers will slightly increase their scores, as the authors complement tasks like classification and regression and demonstrated the effectiveness. However, it seems that some concerns about the validation of new backbone models are not well addressed, where the authors provided one new backbone models and in some cases, it is not better. The performance's concern are also held by the third reviewer.

---

### Decision · Program_Chairs · 2026-01-26

Reject